# YAP charge patterning mediates signal integration through transcriptional co-condensates

Kirstin Meyer [1,2,6], Klaus Yserentant [3,6], Rasmi Cheloor-Kovilakam [3,6], Kiersten M. Ruff [4], Chan-I Chung[1,3], Xiaokun Shu [1,3], Bo Huang [2,3,5] ✉ & Orion D. Weiner [1,2] ✉

Transcription factor dynamics are used to selectively engage gene regulatory programs. Biomolecular condensates have emerged as an attractive signaling module in this process, but the underlying mechanisms are not well-understood. Here, we probe the molecular basis of YAP signal integration through transcriptional condensates. Leveraging light-sheet single-molecule imaging and synthetic condensates, we demonstrate charge-mediated co-condensation of the transcriptional regulators YAP and Mediator into transcriptionally active condensates in stem cells. Intrinsically disordered region sequence analysis and YAP protein engineering demonstrate that the signaling specificity of YAP is established, in part, through complementary electrostatic interactions between negatively charged blocks within YAP and positively charged blocks within Mediator. YAP/Mediator co-condensation is counteracted by negative feedback from transcription, driving an adaptive transcriptional response that is well-suited for decoding dynamic inputs. Our work reveals a molecular framework for YAP condensate formation and sheds light on the function of YAP condensates for emergent gene regulatory behavior.

YAP is a transcriptional regulator that controls a large set of gene programs and cellular decisions, including cellular differentiation, proliferation and pluripotency[1–7]. A long-standing question is how cells achieve YAP signaling specificity to selectively engage the appropriate gene regulatory program in the appropriate cellular context. It is clear that downstream targets interpret YAP's concentrations and dynamics for the precise control of gene activation[7]. However, the molecular basis of these complex YAP signal integrations is unknown. The engagement of YAP in transcriptional condensates makes them an attractive decoding module[8,9].

Transcriptional condensates concentrate gene regulatory proteins to control gene activation[10,11]. These condensates primarily arise through reversible, multivalent interactions of intrinsically disordered proteins and/or RNA, for example, through electrostatic or hydrophobic interactions[12–15]. The cooperative and dynamic nature of condensates makes them excellent feedback circuits for the interpretation of signals from transcriptional regulators, such as YAP[16–20]. For example, the threshold response of genes to YAP levels could be established through the switch-like formation of condensates above a saturation concentration. Combined with negative feedback, condensates function as signaling circuits for more complex signal integrations such as decoding temporal YAP inputs, as observed during early embryonic decisions[7]. Here, we pursue a mechanistic understanding of YAP condensate formation focusing on two key questions. How does YAP

[1]Cardiovascular Research Institute, University of California, San Francisco, San Francisco, CA 94158, USA. [2]Department of Biochemistry and Biophysics, University of California, San Francisco, San Francisco, CA 94158, USA. [3]Department of Pharmaceutical Chemistry, University of California, San Francisco, San Francisco, CA 94143, USA. [4]Department of Biomedical Engineering and Center for Biomolecular Condensates, James McKelvey School of Engineering, Washington University in St. Louis, St. Louis, MO 63130, USA. [5]Chan Zuckerberg Biohub San Francisco, San Francisco, CA 94158, USA. [6]These authors contributed equally: Kirstin Meyer, Klaus Yserentant, Rasmi Cheloor-Kovilakam. ✉e-mail: Bo.Huang@ucsf.edu; Orion.Weiner@ucsf.edu

selectively engage transcriptional condensates? And how does YAP interface with the positive and negative feedback circuits that are essential for the proper regulation of its downstream targets?

Leveraging light-sheet single-molecule imaging, synthetic condensates, and IDR grammar analysis, we demonstrate that charge-mediated co-condensation of YAP and the oppositely charged transcriptional scaffold Med1 drives transcriptional activation. We find that the reciprocal positive feedback underlying YAP/Med1 co-partitioning is mediated through distinct negatively charged blocks within the IDR of YAP. This highlights a mechanism of signaling specificity that is different from the binding of YAP to DNA through TEA domain transcription factors (TEADs). We further unravel the contribution of YAP condensates to transcriptional feedback circuits. This provides a framework for understanding complex YAP signal integrations such as the temporal decoding capacity of genes during early embryonic development[7].

## Results

### Protein charge and solubility determine YAP/chromatin interaction

Although YAP has been previously shown to be able to form condensates in cells at endogenous expression levels, the role of YAP condensates in gene regulation is unclear[8,9]. We begin by probing the potential involvement of condensates in YAP-chromatin interaction. We leveraged a light sheet single-molecule imaging approach to detect YAP chromatin binding events in conjunction with well-controlled perturbations of YAP condensates.

To detect YAP-chromatin binding events by light-sheet single-molecule imaging, we re-expressed a Halo-tagged YAP protein in a YAP

KO background and titrated YAP levels to the endogenous YAP expression range of WT cells to ensure physiologically relevant conditions (Fig. S1A). We only consider cells with nuclear YAP levels within the endogenous expression range. We have previously demonstrated the function of nuclear YAP levels and dynamics for gene regulation and cell fate decisions during cellular differentiation of mouse embryonic stem cells (mESCs at 1.5–3 d post differentiation start[7]). Based on this work, we analyze YAP in spontaneously differentiating mESCs at ~1.5–2 d post differentiation. To visualize chromatin-bound YAP molecules, we leverage single-molecule imaging of sparsely labeled (Halo-tag JFX549) cells. Using a 50 ms camera exposure, chromatin-bound YAP molecules appear as immobile spots, while fast-moving diffusive molecules are blurred out (Fig. 1A). Using a fast acquisition rate (150 ms frame interval), we quantify the dwell times of YAP-chromatin interactions (kymograph, Fig. 1B). The bi-exponential shape of the inverse cumulative distribution function of YAP dwell times reveals two dwell-time populations. These are commonly found for transcriptional regulators and represent non-specific and specific chromatin interactions[21–23]. YAP non-specifically interacts with chromatin with an average dwell time of 1.2 s, while its specific chromatin interactions exhibit an average 7 s dwell time (Fig. 1C). The long dwell times can extend up to ~80 s in rare cases (Fig. 1C).

To test the involvement of condensates for YAP-chromatin interaction, we fused the Halo-YAP construct to the highly soluble protein mCherry and re-expressed it in YAP KO mESCs at endogenous levels. In line with previous reports[24], mCherry is sufficient to dissolve synthetic condensates in our experimental system (Fig. S1B,C). Thus, it should specifically interfere with native YAP condensate formation but not direct binding of YAP to chromatin, as compared to our control in

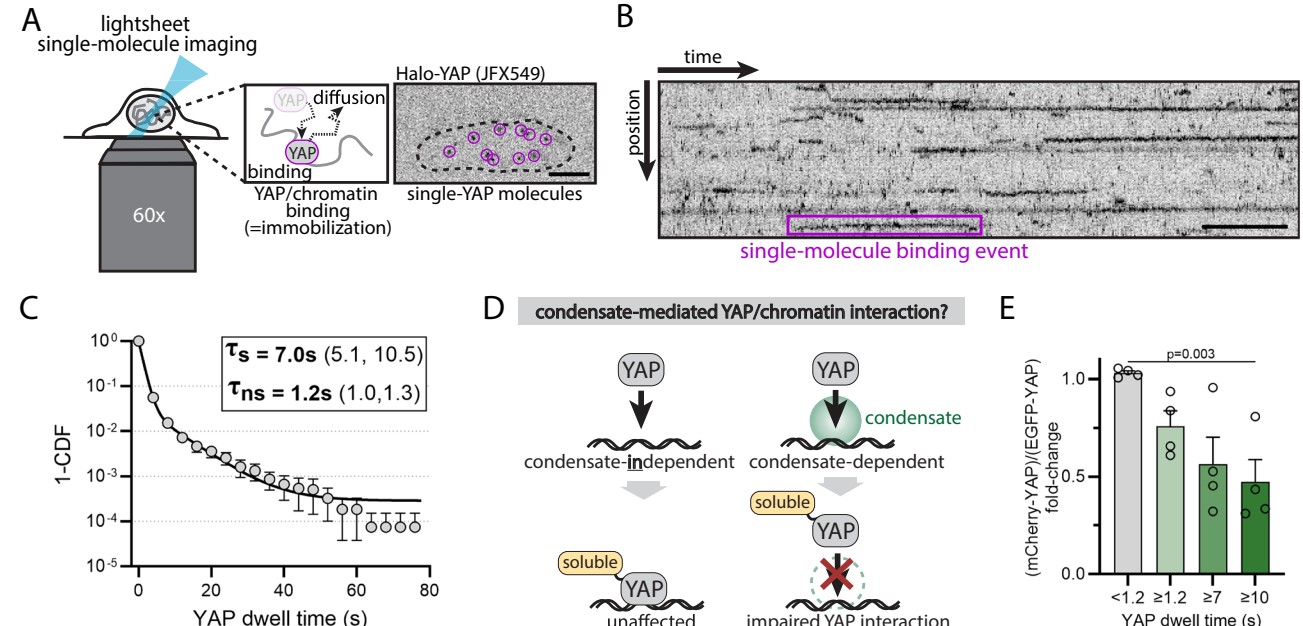

**Fig. 1 | YAP-chromatin dwell times are sensitive to protein solubility, consistent with an involvement of condensates. A** Light sheet single-molecule imaging of sparsely labeled Halo-YAP (JFX549) molecules visualizes chromatin-bound YAP as immobile spots. Unbound YAP molecules exhibit a much higher diffusion constant and are blurred at 50 ms camera exposure times. Image (right) shows single-YAP molecules (outlined by magenta circles) in the nucleus (dashed line) of an mESC. Scale bar: 5 µm. **B** Kymograph of individual YAP-chromatin interaction events in the cell nucleus over time. Scale bar: 10 s. **C** 1-Cumulative Distribution Function (1-CDF) of YAP dwell times quantified from single-molecule time series (as shown in **B**). The curve was fit with a bi-exponential decay function revealing short-lived non-specific ($\tau_{ns}$ = 1.2 s) and long-lived specific ($\tau_s$ = 7 s) YAP dwell times. 95% CI are shown in brackets. Shown are the mean +/– SEM from $N$ = 5 independent experiments. **D** YAP

fusion to a solubility tag that is not expected to affect condensate-independent YAP binding but should impair condensate-dependent YAP interactions. **E** Quantification of YAP dwell times in the presence of a solubility tag (mCherry) or a control tag (EGFP). Shown are the average fold-change for different dwell time populations (unspecific binding events, <1.2 s; specific binding events, >=1.2 s, >=7 s, >=10 s). The solubility tag affects long-lived but not short-lived YAP interactions, consistent with the use of protein condensates for YAP's specific binding events. Shown are mean +/– SEM from $N$ = 4 independent experiments. $p$ values from two-sided unpaired Student's t-test: mCherry-YAP vs EGFP-YAP for dwell times <1.2 s, $p$ = 0.01; dwell times >=1.2 s, $p$ = 0.01; dwell times >=7 s, $p$ = 0.04; dwell times >=10 s; $p$ = 0.03.

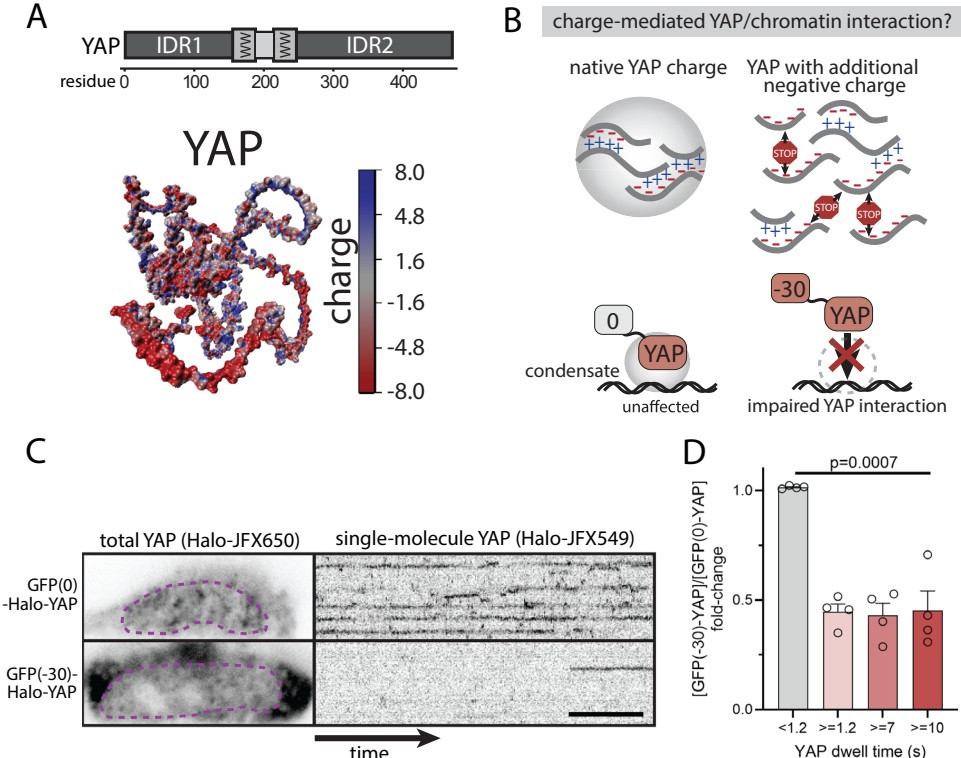

**Fig. 2 | YAP-chromatin dwell times are sensitive to protein charge. A** YAP contains two IDRs (A, top) that are negatively charged (alpha-fold prediction of YAP protein with surface charge annotation, A bottom), raising the possibility of electrostatic interactions for condensate formation. **B** Schematic depiction of YAP's behavior in the presence of a negatively charged tag (net charge: −30), which would be expected to impair electrostatic-based interactions of YAP in condensates. **C** Dual Halo-YAP labeling for simultaneous visualization of total YAP (Halo-tag ligand JFX650) and single YAP molecules (Halo-tag ligand JFX549). Comparing cells with similar total YAP levels (dashed magenta outlines, left panel), the negative charge tag (−30) shows decreased YAP-chromatin interaction events (kymographs, right panel) as compared to a neutral charge tag (GFP(0)). Scale bar, 10 s. **D** Quantitation of YAP single-molecule dwell times from conditions shown in (**C**). Shown are the average fold-change for different dwell time populations (unspecific binding events, <1.2 s; specific binding events, >= 1.2 s, >= 7 s, >= 10 s, as determined in Fig. 1C). Only the specific YAP binding events (>= 1.2 s dwell time) are impaired by the negative charge tag. Shown are mean +/− SEM from $N = 4$ independent experiments. $p$ values from two-sided unpaired Student's t test: GFP(0)-Halo-YAP vs GFP(−30)-Halo-YAP for dwell times <1.2 s, $p = 0.02$; dwell times >= 1.2 s, $p = 0.005$; dwell times >= 7 s, $p = 0.02$; dwell times >= 10 s; $p = 0.004$.

which we replace mCherry with the less soluble EGFP. If condensates play a functional role in YAP-chromatin interaction, solubilization of YAP condensates should affect YAP dwell time distribution (Fig. 1D). Consistent with this expectation, we find a ~ 50% decrease in long-lived YAP dwell times ( > 10 s) in the presence of the solubility tag as compared to the control. In contrast, non-specific YAP-chromatin interactions ( < 1.2 s) remained unaffected (Fig. 1E). These data suggest that at least half of the specific YAP-chromatin interactions are mediated through condensates.

YAP contains two IDRs (Fig. 2A, top) defined by a significant enrichment of negative charge (Fig. 2A, bottom), raising the possibility that condensate formation may involve electrostatic interactions with other oppositely charged partners. To disrupt this mode of condensate formation in the cellular context, we introduced charge imbalance by fusing the Halo-YAP construct with a negative charge-tag (engineered GFP with a net charge of −30)[25] (Fig. 2B). Indeed, the charge-tag phenocopies the effect of the solubility tag, reducing the occurrence of long-lived YAP dwell times by 50% as compared to a neutral tag (engineered GFP with net charge 0), while leaving non-specific interactions unaffected (Fig. 2C, D). While we cannot exclude that the phenotype results from repulsion of the additional negative charge of YAP from the negatively charged phosphate backbone of DNA, our results suggest that long-lived YAP-chromatin interactions likely involve electrostatic interactions of condensates.

## YAP interacts with endogenous transcriptional condensates

One-component macromolecular condensates form when the macromolecular concentration crosses a threshold known as the saturation concentration. Above the saturation concentration, the macromolecular solution separates into coexisting dense and dilute phases. In multi-component systems, the threshold is set either by a single component if homotypic interactions drive condensate formation or jointly by the components whose heterotypic interactions drive condensation[26,27]. To understand YAP condensate formation, we used our expression system to probe the saturation behavior of YAP in mESCs.

Using the same expression system as before (see Fig. 1), we monitored the formation of YAP puncta as a function of nuclear YAP levels in mESCs. When we bracketed the endogenous expression range, light-sheet imaging of fully stained Halo-YAP expressing cells did not reveal apparent macroscopic condensates (Fig. 3A). Instead, we find small and highly dynamic YAP clusters (Fig. 3B) that are only visible at low to intermediate YAP expression levels. Using our single-molecule measurements, we determined the number of YAP molecules per cluster. These clusters barely exceeded ~30 molecules, and larger clusters were very rare (0.01% of clusters have >30 molecules, Fig. 3C). Consistent with this lack of macroscopic clustering, the dilute phase did not show saturation, in which YAP levels exceeding the saturation concentration should drive YAP into condensates while leaving the dilute phase concentration constant. Instead, our results show a steady

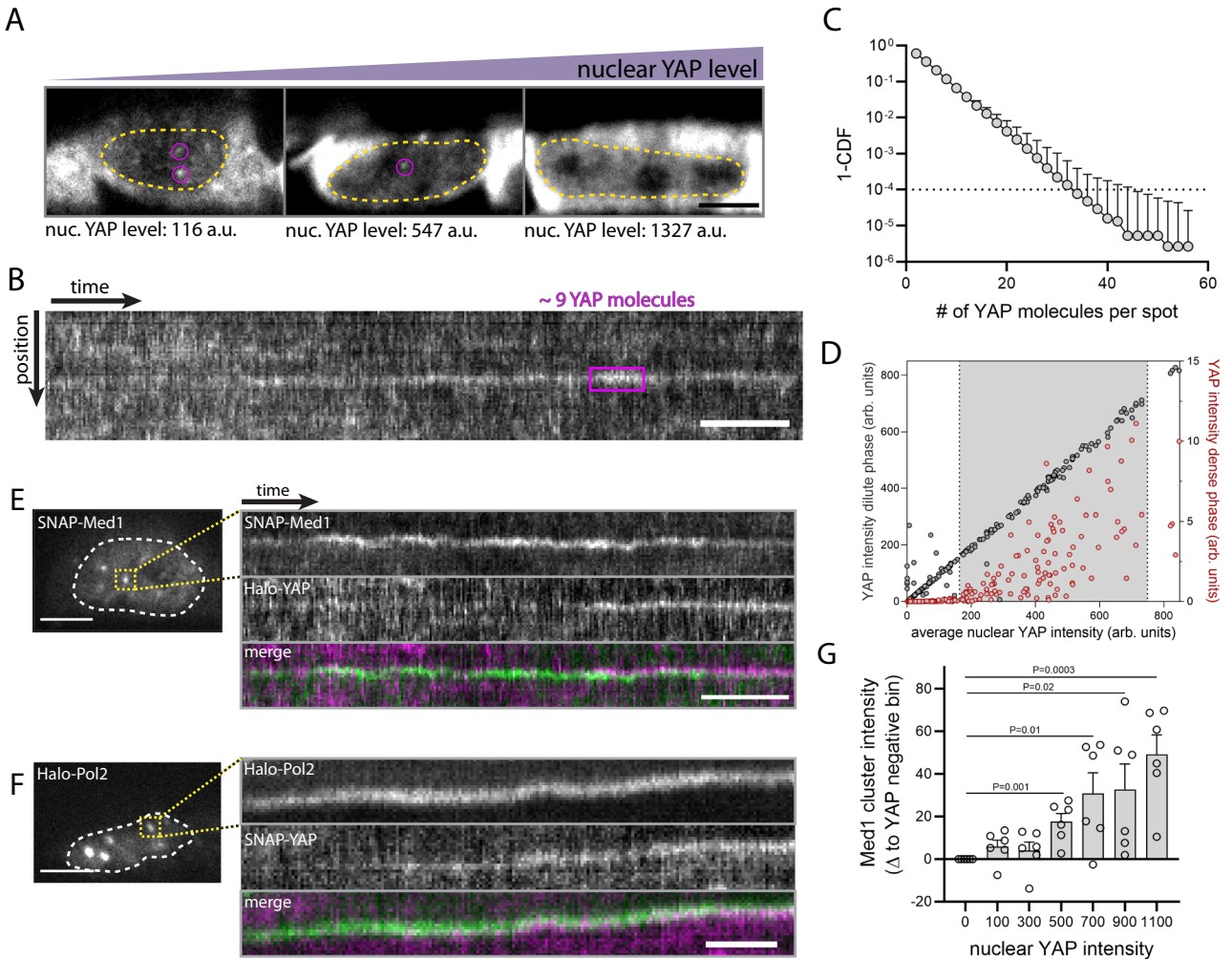

**Fig. 3 | YAP engages with endogenous Med1/Pol2 transcriptional condensates and drives their formation. A** Light-sheet images of total Halo-YAP protein in nuclei with increasing YAP expression levels (left to right). Nuclear clusters (magenta circles) are only visible in nuclei with low/intermediate YAP levels. Nuclei are outlined in yellow. Images are individually min/max scaled to the YAP intensity of each nucleus. Average nuclear YAP levels are indicated. Scale bar: 5 μm. **B** Kymograph of total nuclear Halo-YAP protein reveals YAP clusters containing few molecules, inconsistent with one-component macromolecular condensates. The molecule number per cluster (magenta outline) was estimated from single molecule signals. Scale bar: 10 s . **C** 1-Cumulative Distribution Function (1-CDF) of YAP cluster intensity (expressed as molecule number per spot) quantified from total YAP light-sheet time series as shown in B. Shown are median with upper limit from 5 independent experiments. **D** Quantification of the YAP intensity in the dense phase (right y-axis, integral YAP cluster intensity per nucleus, norm to nuclear area) and soluble phase (left y-axis, average nuclear YAP intensity, excluding YAP clusters) as a function of average nuclear YAP concentration from images as shown in (**A**). Each datapoint represents one nucleus. The gray-shaded background indicates endogenous YAP levels. Shown data is pooled from *N* = 6 independent experiments. The graph is a zoom-in of a larger data range, see Fig. S2. **E**, **F** Simultaneous light sheet microscopy of endogenous SNAP-Med1 or Halo-Pol2 condensates (bright nuclear spots, left) and Halo-YAP or SNAP-YAP, respectively. Med1 and Pol2 condensates were identified as bright nuclear spots (left image). Kymographs (right images) show transient recruitment of YAP to Med1 and Pol2 condensates. Scale bar, 20 s. **G** Quantification of Med1 condensate intensity (integral per nucleus, norm to area) as a function of nuclear YAP levels from maximum projection of confocal image stacks. Shown are mean +/− SEM from *N* = 6 independent experiments. *p* values from two-sided unpaired Student's t test.

increase of dilute phase concentrations under concentrations where YAP clusters occur (Fig. 3D). These results are inconsistent with a condensate system that follows a simple saturation behavior, with saturation set by a single component.

YAP could follow a non-stereotypical saturation behavior if it is part of a more complex co-condensation system. In this case, the interaction of co-condensate components determines the partitioning behavior[26]. To test this hypothesis, we set out to visualize possible condensation partners under physiological conditions in mESCs. The Mediator complex is one of the main components of transcriptional condensates and is also known to interact with YAP[28–30]. The Mediator complex contains a large number of subunits of which Med1 harbors a main IDR and is sufficient for condensate formation in vitro[31,32]. We used CRISPR/Cas9 to endogenously tag the Med1 subunit in our Halo-

YAP mESCs. Time-lapse light-sheet imaging of Med1 reveals long-lived Med1-containing nuclear condensates, as previously described[31]. Two-color imaging of Med1 and YAP reveals transient accumulation of YAP at Med1 puncta on the timescale of seconds (Fig. 3E). These data suggest that Med1 recruits and/or co-condenses with YAP. Mediator condensates have previously been characterized to represent transcriptionally active compartments by recruiting phase-separated RNA Polymerase 2 (Pol2)[31]. To test if YAP also interacts with Pol2-positive condensates, we used the same CRISPR/Cas9 strategy to visualize the endogenous Pol2 complex by tagging the IDR containing subunit RPB1. Simultaneous light-sheet imaging also reveals a transient accumulation of YAP at Pol2 puncta, demonstrating that YAP is a component of transcriptionally active compartments (Fig. 3F). Together, these data reveal that Med1/Pol2 condensates can recruit YAP and suggest that

YAP and Med1 likely undergo co-phase separation to form condensates.

In a two-component system, co-condensation is driven by a blend of homotypic and heterotypic interactions. For a given expression level of the two components, the threshold for co-condensation is crossed when component 1 crosses a threshold $c_1$ and component 2 crosses a threshold $c_2$. Together, the geometric mean of these concentrations defines a solubility product or solubility threshold[27], and the precise value of this threshold is determined by the shape of the phase boundary, which is in turn determined by the blend of heterotypic and homotypic interactions[16,26]. The presence of one component influences the threshold concentration of the other. If there is positive cooperativity[16,26], then we would expect enhanced partitioning of both components in the condensed phase. Indeed, when we quantified the intensity of Med1 puncta as a function of YAP expression levels using confocal microscopy, we observed a significant increase of Med1 condensates with increasing YAP levels (Fig. 3G). To further test the co-condensate model and the reciprocity of YAP-Med1 interaction, we used a synthetic condensate system (SPARK-ON,[33]) that enables us to acutely reconstitute YAP condensates in mESCs. The SPARK-ON system leverages a small molecule (lenalidomide) to induce the dimerization of two synthetic multimerization domains (HOTag3, HOTag6, Fig. 4A). This added valency drives nuclear condensate formation, providing a functionally inert condensate scaffold. The addition of the rapamycin dimerizing proteins (Frb/FKBP) on the scaffold (Frb) and YAP (FKBP) enables the rapamycin-inducible recruitment of YAP to the synthetic condensates. The rapamycin-mediated recruitment is acute and occurs within 1-2 min[33]. Using confocal imaging we first tested if YAP alone is sufficient to recruit Med1 to condensates. We transiently transfected endogenous SNAP-Med1 reporter lines in the YAP KO background with the SPARK-ON system and re-expressed Halo-YAP as a fusion protein with FKBP. To reach a steady-state of the synthetic condensate scaffold, we incubated mESCs with lenalidomide for 30 min prior to rapamycin-induced YAP recruitment. Leveraging time-lapse imaging, we monitored the response of Med1 to acute YAP recruitment to the synthetic condensates (Fig. 4B,C). YAP instantaneously recruits Med1 to the synthetic condensates (Fig. 4C) where it remains present over the time course of the experiment (2 h). Together with our observations of the endogenous system (see Fig. 3E), these data demonstrate that Med1 and YAP recruit one another, consistent with a co-condensation system.

Co-condensation enables YAP-Med1 interaction to produce an apparent positive feedback effect in the transcriptional response to YAP signaling. We postulate that YAP and Med1 co-condense through electrostatic interactions of their oppositely charged IDRs (YAP, negatively charged; Med1, positively charged, Fig. 4D, top) based on our earlier observation of charge-sensitivity of YAP dwell time (Fig. 2). To test this effect, we interfered with the electrostatic interaction of the YAP-Med1 complex. To this end, we expressed FKBP-YAP fused to the negative charge tag in our SPARK-ON system (Fig. 4D, bottom). Comparing synthetic condensates with similar rapamycin-induced YAP recruitment levels (Fig. 4E, compare red curves at the dashed line), we observe a complete loss of the Med1 response for the charge-tagged YAP protein as compared to control YAP (Fig. 4E, compare blue curves). Correlated with the lack of Med1 recruitment to synthetic condensates, the charged-tagged YAP construct did not further amplify its own recruitment relative to the control (Fig. 4E, compared red curves). These results demonstrate a positive charge-mediated feedback cycle between YAP and Med1 that drives their co-partitioning.

## YAP/Med1 co-condensates drive transcriptional activation

To test the physiological function of the YAP/Med1 co-condensates, we recorded the response of our endogenous Pol2 reporter line to acute YAP condensate formation with the SPARK-ON system. Using the same conditions as before (see Fig. 4A–C), we acutely formed YAP condensates and monitored the endogenous Pol2 response (Fig. 5A, B). YAP condensates suffice to recruit Pol2. In contrast to the sustained recruitment of Med1, the Pol2 recruitment is adaptive following sustained YAP condensate formation. This observation is consistent with a negative feedback circuit based on transcriptional activation and is reminiscent of RNA-mediated negative feedback reported previously[18]. Negative feedback is an important feature of dynamic signal decoders. Following an initial stimulus, the negative feedback resets the system to baseline, effectively acting as a change detector of pulsatile inputs, as previously shown for YAP[7]. Given the importance of delayed negative feedback for such complex signal integrations, we probed its molecular basis. To this end, we leveraged the Pol2 inhibitor DRB (5,6-Dichloro-1-β-D-ribofuranosylbenzimidazole). DRB blocks the elongation of RNA synthesis by inhibiting CDK9 but does not directly affect Pol2 recruitment to gene loci[34] (Fig. 5C). We added the Pol2 inhibitor at 10 min post YAP recruitment, when the Pol2 response was approaching its maximum (see Fig. 5D). Consistent with negative feedback originating from the transcriptional output, DRB treatment converted the adaptive response into sustained Pol2 recruitment (Fig. 5D). The phenotype was only detectable in condensates with low levels of YAP recruitment, suggesting that YAP and RNA, which are both negatively charged components, may act in concert to drive the negative feedback. Interestingly, YAP intensity at the condensates also showed a significant increase in YAP recruitment following DRB treatment (Fig. S3). Because both conditions received the same amount of rapamycin to recruit YAP, these data indicate that transcription-based negative feedback also controls the recruitment of YAP itself. Together, we reveal an entire feedback cycle of YAP-mediated transcriptional condensates. YAP potentiates its own recruitment through co-condensation with Med1 but is counteracted by delayed negative feedback from the transcriptional output.

To probe the functional role of YAP-induced transcriptional condensates, we analyzed differentially expressed genes by RNA-seq following synthetic YAP condensate formation. Using our synthetic condensate system (SPARK-ON, see Fig. 4A), we compared RNA transcripts in cells with rapamycin-induced YAP recruitment to synthetic condensates versus cells with comparable YAP expression but without condensate recruitment. We chose to analyze gene expression at 1 h post YAP condensate recruitment based on our previous observations that YAP-induced Pol2 recruitment peaks around this time (see Fig. 5A, B). Interestingly, despite the presence of Pol2 at the condensates, the differential gene expression analysis reveals a repressive effect of YAP condensates on gene expression. Out of ~20,000 detected transcripts, 33 are significantly reduced, while only one shows increased levels (Fig. S6A, B). While we cannot exclude that the decrease of gene expression is caused by the strong sequestration of Pol2 to the condensates at the expense of Pol2 levels at other genes, the lack of transcript level induction is at odds with the presence of Pol2 at the condensates. This may point towards a role of Pol2 for expression of regulatory RNAs (e.g. enhancer RNAs) that play an important role of Mediator condensate assembly but are not captured by our RNAseq approach.

## YAP charge patterning mediates Med1 co-condensation

The multivalent interactions that form the basis of condensate formation can be established through a number of reversible physical cross-links, including electrostatic and hydrophobic interactions[12,13,35,36]. Our results suggest that electrostatic interactions of YAP and Med1 drive their co-partitioning. How do these physical crosslinks establish interaction specificity? Previous reports have demonstrated specificity through the patterning of cohesive motifs within IDRs, including the distribution of charged residues along the protein sequence (blockiness vs dispersion)[32,36,37]. Here, we set out to test if the patterning of non-random sequence features of the YAP, Med1, and Pol2 IDRs can explain

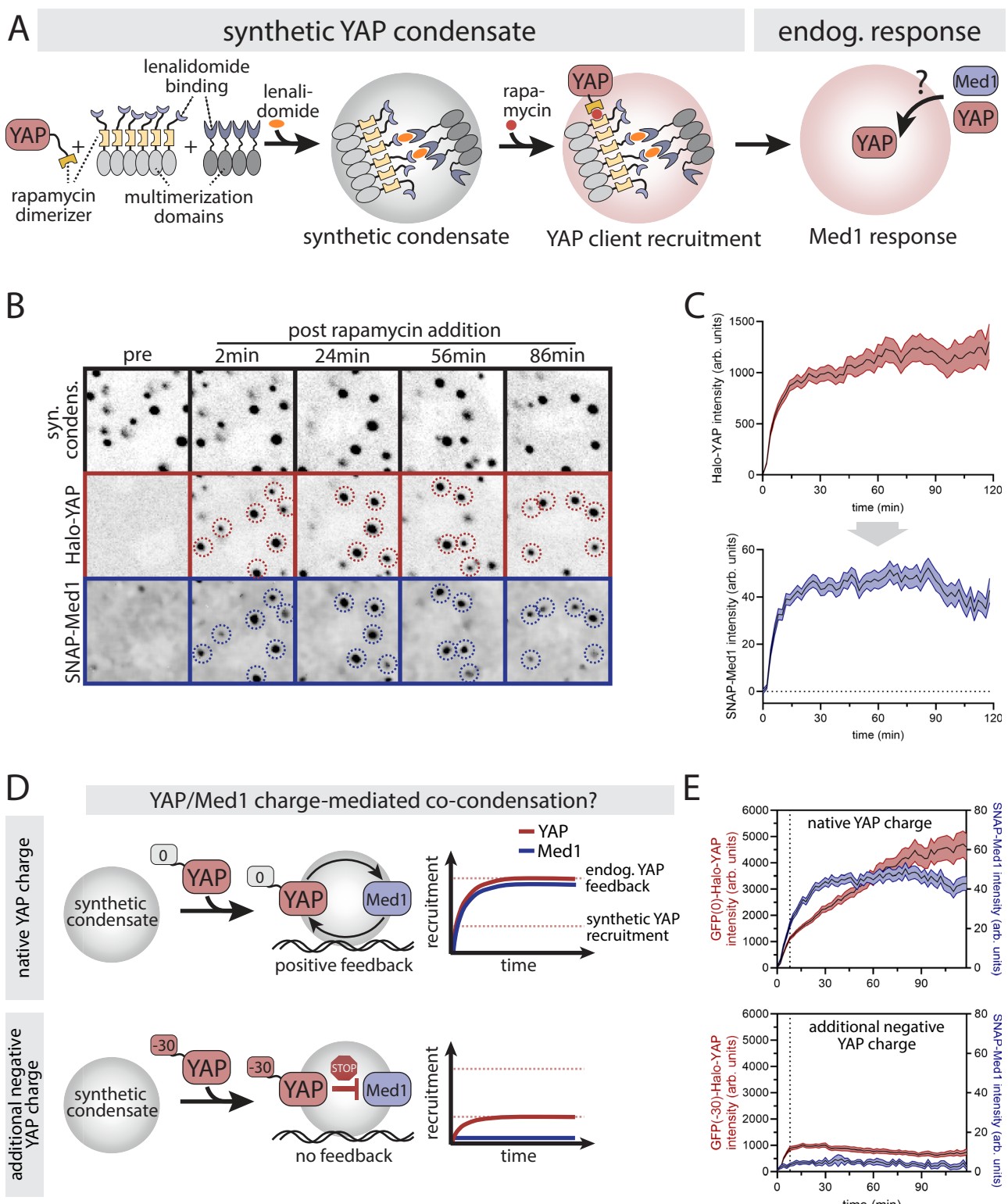

the basis of their specific interactions. While the Mediator and Pol2 complex are large multi-subunit complexes, only a few subunits contain IDRs. We focused on the IDR of the Mediator1 and RPB1, which have previously been shown to be sufficient for condensate formation in vitro[32,38]. To probe the contribution of charge relative to other sequence parameters, we leveraged a previously-reported computational approach, NARDINI+, that considers a comprehensive set of 90 different IDR sequence features that are relevant for condensate formation[39]. These include the enrichment of individual amino acids and

the patterning of distinct pairs of residue types with respect to one another. The approach tests for the significant enrichment or depletion of compositional biases in the IDR of interest compared to all IDRs in the mouse genome, as well as the degree of patterning observed compared to compositionally identical random sequences. Our analysis reveals a significant enrichment in the segregation of negatively charged residues, and the segregation of negatively charged residues with respect to polar residues in the C-terminal IDR of YAP (Fig. 6A, B). In contrast, the N-terminal IDR of YAP shows an enrichment in Pro and Ala rich stretches

**Fig. 4 | YAP forms co-condensates with Med1 through charge-mediated positive feedback. A** Synthetic condensate system for probing the function of YAP for Med1 condensate formation: the lenalidomide-inducible interaction of two engineered multimerization domains drives the formation of a synthetic condensate.[33] A separate rapamycin-based chemical dimerizer system is used to recruit YAP to these condensates. The resulting endogenous Med1 and YAP response provides information on Med1/YAP co-condensation and feedback. **B** Time series images of acute YAP recruitment (middle row) to pre-formed synthetic condensates (top row) and the resulting response of the endogenous Med1 protein. YAP was recruited to synthetic condensates at 2 min following rapamycin addition. Blue and red dashed circles indicate Med1/YAP double positive condensates. **C** Quantification of time series as shown in B. Top (input): YAP recruitment after addition of rapamycin at 0-2 min. Bottom (output): Med1 recruitment to the synthetic condensates. Shown are mean +/− SEM from pooled time courses of 4 independent experiments. **D** Schematic depiction of YAP/Med co-condensation behavior in the presence of a negatively charged tag (net charge: −30), which interferes with YAP's positive feedback. Top: Synthetic YAP recruitment drives a positive feedback cycle where electrostatic interactions of YAP (negatively charged) and Med1 (positively charged) drive YAP/Med1 co-condensation. The system is unaffected by a neutral charge tag (0). Bottom: Interference of YAP's native charge through the addition of a negative charge tag (net charge: −30) which impairs the electrostatic interaction with Med1. This inhibits both Med1 recruitment as well as the positive feedback required for YAP/Med1 co-condensation. **E** Quantification of YAP (top) and Med1 (bottom) recruitment to acute formation of synthetic YAP condensates, comparing YAP fused to a neutral (GFP(0)) with a negative-charged (GFP(−30)) tag. Despite comparable initial rapamycin mediated-YAP recruitment (dashed line, compared red curves in top and bottom graph), the additional negative charge inhibits the positive feedback required for YAP/Med1 co-condensation (blue curve, bottom graph), as compared to the control (blue curve, top graph). Shown are mean +/− SEM from pooled time courses of $N = 6$ independent experiments.

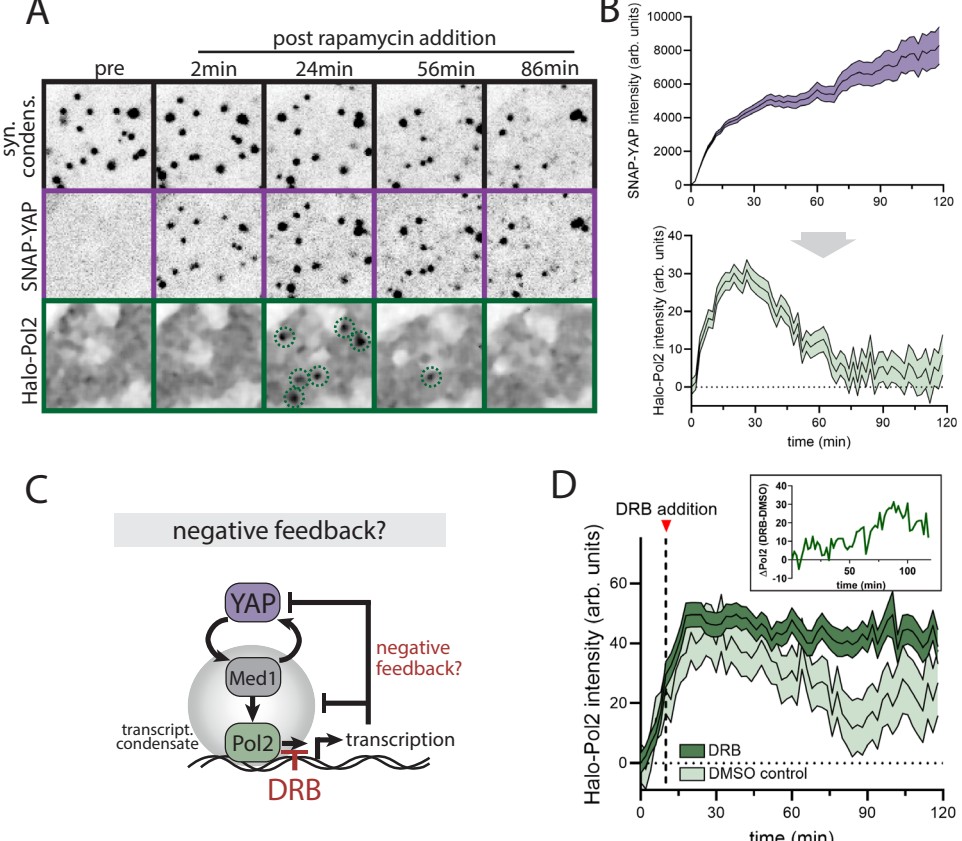

**Fig. 5 | Transcriptional activation drives negative feedback of YAP/Med1 co-condensates. A** Time series images of acute YAP recruitment (middle row) to pre-formed synthetic condensates (top row) and the resulting response of the endogenous Pol2 protein. YAP was recruited to synthetic condensates at 2 min following rapamycin addition. Green dashed circles indicate transient Pol2 positive condensates. **B** Quantitation of time series as shown in A. Top (input): YAP recruitment after addition of rapamycin at 0-2 min. Bottom (output): Pol2 adaptive recruitment to the synthetic condensates. Shown are mean +/− SEM from pooled time courses of $N = 4$ independent experiments. **C** Testing the transcription-mediated negative feedback circuit: YAP/Med1 co-condensation initiates Pol2 recruitment. Following transcriptional activation, the output inhibits YAP and Pol2 recruitment (negative feedback). The transcriptional inhibitor DRB blocks transcriptional elongation and negative feedback. **D** Quantitation of Pol2 recruitment upon YAP condensate formation following inhibition of RNA synthesis with DRB at $t = 12$ min (dashed line). The addition of DMSO serves as control. Inset shows the difference in the Pol2 recruitment of DRB-treated cells to DMSO control cells. Shown are mean +/− SEM from pooled time courses of $N = 4$ independent experiments.

as well as an enrichment in hydrophobicity. Consistent with previous reports, we detect enriched segregation of positively charged, polar and hydrophobic residues in the Med1 IDR[32]. Together, the presence of blocks of acidic groups in the C-terminal IDR of YAP and basic groups in the IDR of Med1 are suggestive of complementarity that could modulate the specificity of their interactions. In contrast, the Pol2 IDR shows distinct features (enriched in well-mixed aromatic, polar, and proline residues) from YAP and Med1, suggesting that Pol2 may not be directly recruited through YAP itself or could rely on different interactions.

Previous reports have highlighted the importance of charge patterning for selective condensate interactions in vitro[32,36,40]. The complementary charge patches of YAP and Med1 (see Fig. 6B) suggest that

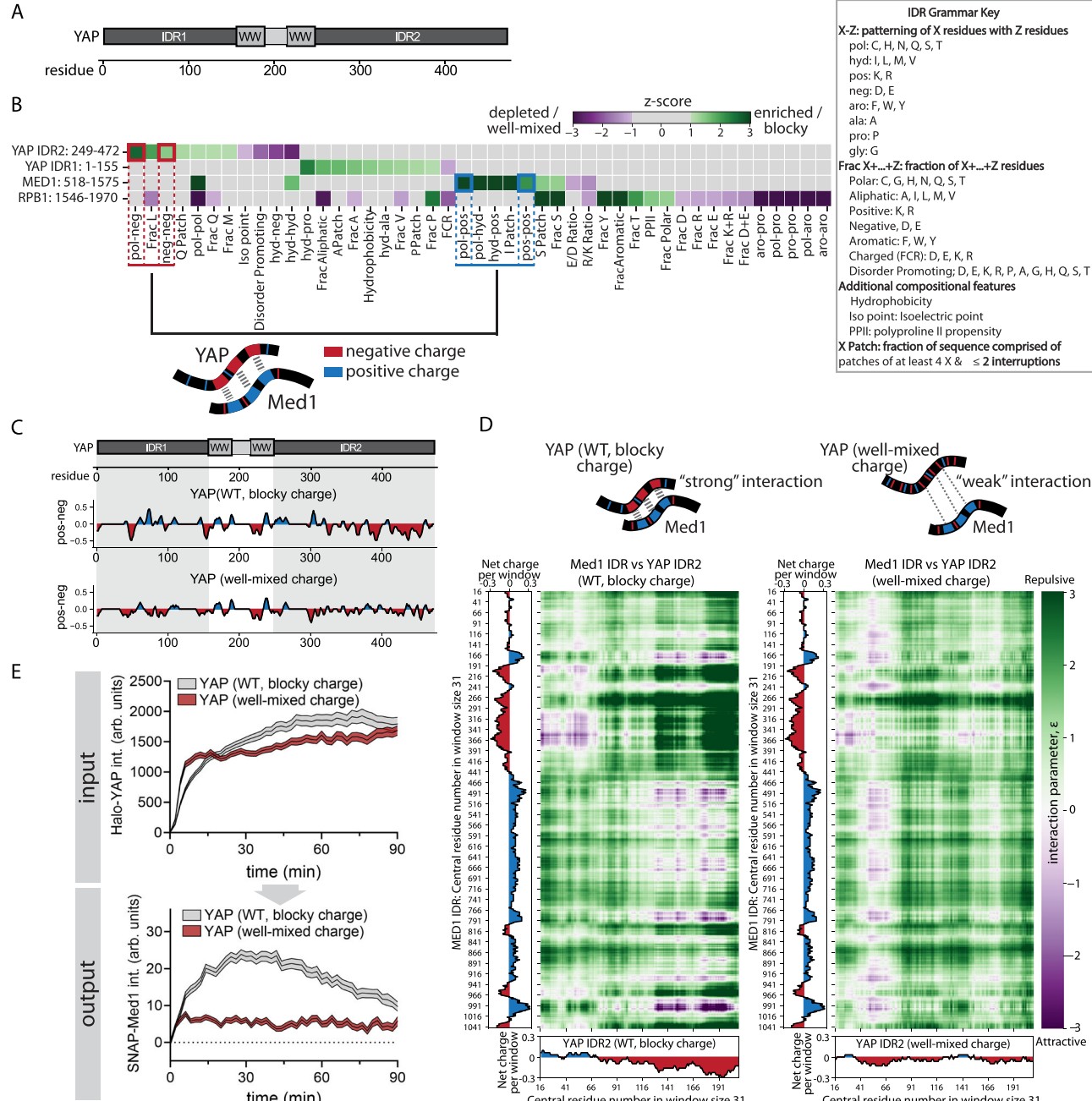

**Fig. 6 | YAP charge patterning drives YAP/Med1 co-condensation. A** YAP protein contains 2 IDRs (IDR1, residues1-155; IDR2, residues 249-472), which are separated by two WW domains. **B** NARDINI+ analysis of sequence features across the YAP IDRs 1 and 2, the Med1 IDR and the Pol2 subunit RPB1 IDR. See IDR Grammar Key (right) for features analyzed that showed exceptional grammars. YAP and Med1 are significantly enriched for negative and positive charge blocks, respectively. Note that RPB1 is not significantly enriched in charge-based features. **C** YAP sequence engineering to test the requirement of negative charge blocks in YAP IDR2 (as opposed to overall charge of YAP) for Med1 recruitment. The blocks of negative charge in the YAP IDRs (WT protein, blocky charge, top panel) were dispersed throughout both IDRs (well-mixed charge construct, bottom panel) while maintaining the total protein net charge. Graph shows the mean net charge per residue along the sequence of the WT YAP protein (top graph) and the well-mixed charge variant (bottom graph). The structured domains (WW domain) were left unchanged. **D** Predicted intermolecular interaction maps for the Med1 IDR and YAP IDR2 of the WT (blocky charge pattern, left map) and engineered well-mixed charge YAP

variant (right map).[42] Attractive (magenta) and repulsive (green) interaction strengths are predicted based on the chemical physics of the Mpipi force field. Note the alignment of the precited interactions with the IDR charge patterns of Med1 (left graph) and native YAP IDR2 (bottom graph) but weaker interactions in the well-mixed YAP charge construct. **E** Med1 recruitment to acute YAP condensate formation using the synthetic condensate system (see Fig. 4A), comparing the WT blocky-charge protein with the engineered well-mixed charge variant (see **C**). Top (input): Recruitment of Halo-YAP variants (WT blocky charge vs well-mixed charge) to synthetic condensates. Bottom (output): Endogenous SNAP-Med1 response to the acute YAP condensate recruitment (top graph). SNAP-Med1 and Halo-YAP variants were imaged simultaneously. Shown are mean +/− SEM of pooled time-series from *N* = 5 independent experiments. Wildtype YAP (blocky negative charge distribution) efficiently recruits Med1, but well-mixed YAP fails to do so, indicating that the negative charge pattern of YAP determines its specific interaction with Med1.

charge patterns may mediate their specific electrostatic interactions. Alternatively, just net charge alone might suffice as a driver of YAP-Med1 interactions. This would be reminiscent of a mean-field electro-statics view of phase separation[41]. To distinguish between these possibilities, we designed and engineered a YAP variant with dispersed negatively charged residues (Fig. 6C). This design maintains the net charge of the endogenous IDRs but disperses the negatively charged residues throughout the sequence (Fig. S4A). The resulting charge dispersion is well-mixed compared to random sequences with the same composition. Importantly, the positioning and the structures of the WW domains remained unaltered. Furthermore, all non-charge related IDR sequence features of the engineered YAP variant were maintained to be equivalent to the endogenous YAP sequence. To test the effect of charge dispersion on the interaction between YAP IDR2 and Med1 IDR, we used computations based on the FINCHES program to identify putative regions that engage in complementary attractive and antagonistic repulsive intermolecular interactions[42]. FINCHES computes a mean-field interaction parameter between all unique residue windows of two IDRs. The resulting interaction map quantifies the relative strengths of attractions and repulsions for windows of residues along the IDRs (Fig. 6D, YAP WT, left map). Aligning the charge pattern with the interaction map shows that attractive inter-actions largely follow the distribution of the opposite charge patterns between Med1 IDR and YAP WT IDR2. These data support the hypothesis that the complementary charge patterns are likely to be the key determinants of heterotypic interactions between Med1 and YAP. The dispersion of charged residues in the well-mixed YAP variant reduces the predicted interaction strength with Med1 (Fig. 6D, YAP well-mixed charge, right map), underscoring the importance of mul-tivalent complementary electrostatic interactions.

To test these predictions, we used our synthetic YAP condensate system and queried the ability of the well-mixed YAP charge variant to recruit endogenous Med1 to condensates. Quantification of the Med1 response to acute YAP recruitment reveals significantly impaired Med1 recruitment in response to the well-mixed charge YAP variant (Fig. 6E). While the WT YAP protein recruited Med1 within minutes, the Med1 response to the well-mixed charge variant was significantly reduced. These data suggest that the key determinants of specificity are the complementary charge patterns within the YAP IDR2 and the Med1 IDR. We further tested if the importance of YAP charge patterning propagates throughout the transcription initiation cascade by quan-tifying the response of Pol2 to the well-mixed YAP variant. The absence of Med1 also causes a lack of Pol2 recruitment (Fig. S4B), and this is consistent with the function of Med1 as an upstream regulator of Pol2. We leveraged the intermolecular interaction analysis to probe if the impaired Pol2 recruitment is a direct consequence of the altered YAP charge pattern or is a secondary consequence of impaired Med1 interaction. The YAP IDR2 is the only part of the protein that shows a predicted interaction with the Pol2 IDR (Fig. S4C). The well-mixed charge YAP variant is predicted to have a slightly enhanced interaction with the Pol2 IDR (decrease epsilon, see Fig. S4C), which contrasts with the experimentally-observed decrease in Pol2 interaction for this YAP variant (Fig. S4B). These data suggest that YAP charge patterning specifically mediates its interaction with Med1, while Pol2 is recruited downstream of Med1.

The Mediator complex is a large multi-subunit complex, out of which five subunits (Med1, Med14, Med15, Med25, Med26) contain large IDRs ($\geq 100$ residues; Fig. 5SA)[31,43]. While our data demonstrates a charge-mediated interaction of YAP and Med1, we cannot rule out that YAP also interacts with other subunits to initiate the assembly of transcriptional condensates. To test if synthetic YAP condensate for-mation also recruits other components of the Mediator complex, we used CRISPR/Cas9 to label endogenous Med15 with a SNAP tag and repeated our synthetic YAP condensate recruitment assay (Fig. S5B). Med15 has been shown to form condensates and is implicated in

transcriptional initiation[44]. Similar to Med1, rapamycin-induced YAP condensate formation recruits Med15 within few minutes (Fig. S5B). Over the time course of 2 h, the Med15 signal stays sustained, indi-cating lack of negative feedback from the transcription machinery as observed for Pol2 (see Fig. 5B). This suggests that YAP-Med1 co-con-densation is accompanied by recruitment of other Meditator subunits such as Med15. To further probe the role of YAP's charge pattern for recruitment of other Mediator subunits, we computationally analyzed the mean net charge distribution per residue along the IDRs of Med1, Med14, Med15, Med25, and Med26 (Fig. S5C). Among these, Med26 is the only IDR other than Med1 with strongly charged regions, exhibiting positively charged patches. Analysis of the predicted interactions between the YAP IDRs and the Med26 IDR shows attractive interaction scores that coincide with YAP IDR2's negatively charged patch (Fig. S5D). This suggests that YAP's charge pattern may not only drive co-condensation with Med1 but also Med26. Interestingly, we also observed strong predicted interactions between YAP's IDR2 and the IDR of Med15 (Fig. S5E). However, these did not follow the logic of YAP's charge profile. Instead, they are consistent with the localization of the Q-rich regions of Med15 and the polar regions of YAP's IDR2, suggesting an involvement of these residues in YAP-Med15 interac-tions. These results suggest that, in addition to its charge-mediated co-condensation with Med1, YAP may play a more central role in Mediator complex formation through alternate interactions of its glutamine-rich or charged regions with Med15 or Med26, respectively.

Together, our results reveal the physico-chemical mechanisms of YAP/Med1 co-partitioning. These interactions establish the positive and negative feedback cascades that are crucial for establishing the dynamics of transcriptional activation (Fig. 7). We demonstrate the essential role of YAP charge patterning for its specific interaction with Med1 through electrostatic interactions. Here, the increase of YAP protein levels drives its cooperative co-condensation with Med1 and the recruitment of the transcription machinery (Fig. 7a, b). The delayed transcriptional activation drives the negative feedback that terminates the transcriptional response. As a result, changes in YAP levels create an adaptive transcription cycle (Fig. 7c) that might explain emergent gene regulatory behavior such as the decoding of temporal signaling inputs[7]. Based on our computational interaction analysis, it is likely that this mechanism also involves other Mediator subunits such as Med26.

## Discussion

YAP has the capacity to form condensates that represent attractive signaling modules for emergent gene regulatory behavior[8,9]. Yet, how YAP signals are integrated through condensates to control transcrip-tional dynamics remained unclear. Here, we leveraged light-sheet single-molecule imaging, synthetic condensates, and IDR sequence analysis to probe the formation and function of YAP condensates. The formation of these condensates is facilitated by co-condensation of YAP and the transcriptional regulator Med1 (Figs. 3 and 4). The pat-terning of charged residues within the IDR2 of YAP appears to be a key determinant of specificity for the Med1 interaction that drives the formation of these condensates and downstream transcriptional acti-vation (Fig. 6). Condensate growth can be counteracted by delayed negative feedback from the transcriptional output, generating an adaptive transcriptional response (Fig. 5). Our work provides a potential molecular basis for how downstream targets are activated in response to a specific threshold or temporal dynamics of YAP activation.

It has previously been shown that cells leverage YAP levels and dynamics to differentially engage gene programs in control of pro-liferation, differentiation, and pluripotency[7]. Here, genes detect YAP concentrations through switch-like threshold responses, while pulsa-tile inputs can be decoded through adaptive transcriptional responses. However, the decoding mechanisms underlying these preferential

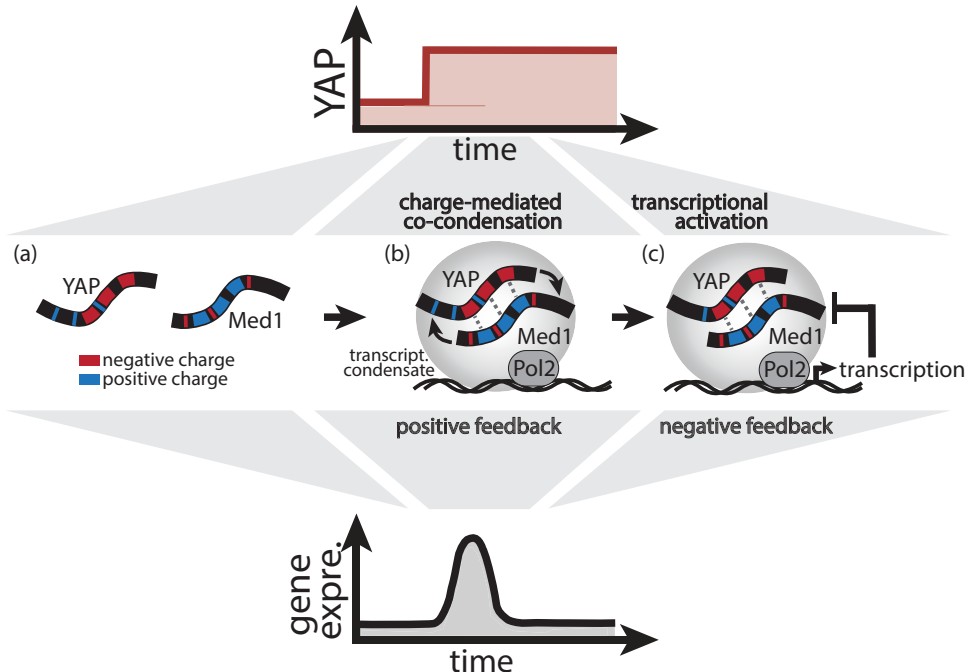

**Fig. 7 | YAP signal integration through Med1 co-condensation and transcriptional feedback.** Proposed model for YAP signal integration through cooperative co-condensation and negative feedback of transcriptional condensates: following an increase of nuclear YAP levels (top graph), complementary multivalent electrostatics interactions of oppositely charged patches of YAP and Med1, respectively (**a**), drive switch-like co-condensation (positive feedback, **b**). Concomitant recruitment of Pol2 results in transient transcriptional activation (panel b) followed by negative feedback and transcriptional adaptation (**c**).

responses have remained unclear. Our results suggest that the cooperative nature of YAP/Med1 co-condensation sets sharp boundaries for condensate formation at a threshold level of YAP. Combined with the adaptive nature of the transcriptional response, these behaviors establish the foundation for temporal decoders. If the acute increase of YAP favors YAP/Med1 co-condensation, the system enables frequency decoding of pulsatile YAP inputs, as would be the case during embryonic stem cell differentiation[7]. Interestingly, our results demonstrate that YAP and Pol2 are more sensitive to negative feedback from the transcriptional output than Med1. This raises the possibility that once YAP/Med1 co-condensates are formed, YAP is dispensable for compartment maintenance but leverages Med1 as a template for continuous cycles of YAP/Pol2 recruitment. This is supported by our observations of endogenous Med1 condensates which show transient recruitment of YAP to stable Med1 compartments. As such, Med1 could act as a memory component that renders genes competent for multiple YAP/Pol2 transcription cycles.

Of note, the observed YAP phase behavior of our study differs from previous reports that found large macromolecular YAP assemblies and an involvement of hydrophobic interactions of YAP's coiled coil domain for YAP condensation[9,45]. Importantly, the YAP isoform used in our study, the one that predominantly underlies YAP's role of mESCs physiology, lacks a part of the coiled-coil (exon 6)[46]. This may explain why electrostatic interactions of the IDRs dominate YAP co-condensation behavior in our system and indicate that condensation mechanisms may be controlled through alternative splicing.

Beyond YAP target regulation, it is likely that the principles we have uncovered here are broadly applicable to how cells establish threshold responses and dynamic decoding of other transcriptional regulators, including p53, NFkB, and Erk[47–50]. Towards this end, it will be interesting to probe whether these transcriptional regulators engage through similar condensate-dependent mechanisms to control the dynamics of the transcription machinery. Importantly, co-condensation mechanisms, as observed for YAP, can involve small

numbers of individual components (YAP molecules). Our observations were made possible through light-sheet imaging that provides sufficient sensitivity to resolve small YAP clusters. It is likely that other regulators act in similar ways but that are missed using common imaging modalities (e.g., spinning disk confocal microscopy).

Our data demonstrate negative feedback from the transcriptional output on YAP/Med1 co-condensates, but the basis of this negative feedback is not known. We find that this negative feedback originates downstream of transcriptional elongation. The key output of transcription could be the formation of RNA; previous work established RNA-mediated feedback that dissolves Med1 condensates at high concentrations in vitro[18]. However, in our system (mESCs), the negative feedback from transcriptional activation acts on YAP and Pol2 but leaves the Med1 compartment intact, suggesting that the negative feedback could be more complicated than the accumulation of RNA. Visualizing the dynamics of RNA accumulation and actively manipulating RNA features (charge, base pair composition) could provide an answer.

## Methods

### mESC culture maintenance and spontaneous differentiation
E14 mESCs (gift from the Panning lab, UCSF) were maintained on gelatin coated dishes in 2i+LIF media, composed of a 1:1 mixture of DMEM/F12 (Thermo Fisher, 11320-033) and Neurobasal (Thermo Fisher, 21103-049) supplemented with N2 supplement (Thermo Fisher, 17502-048), B27 with retinoid acid (Thermo Fisher, 17504-044), 0.05 % BSA (Thermo Fisher, 15260–037), 2 mM GlutaMax (Thermo Fisher, 35050-061), 150 μM 1-thioglycerol (Sigma, M6145), 1 μM PD0325901O (Selleckchem, 1036), 3 μM CHIR99021 (Selleckchem, S2924) and 10⁶ U/L leukemia inhibitory factor (Peprotech, 250-02).

For spontaneous differentiations, cells were spun out of the 2i+LIF media and seeded in spontaneous differentiation media composed of DMEM high glucose (Thermo Fisher Scientific, 11995-073), 15 % ES-qualified FBS (Thermo Fisher, 16141079), 2 mM L-Glutamine (Gibco,

35050061), 0.1 mM non-essential amino acids (Gibco, 11140-050), and 150 μM thioglycerol (Sigma Aldrich, M6145). For seeding conditions, see below.

### CRISPR/Cas9 editing and cell line generation

For the generation of mESC reporter lines in the YAP KO background[7], we used the sgRNA/Cas9 dual expression plasmid pX330 (gift from Feng Zhang, Addgene #42230)[51] and inserted a sgRNA coding sequence targeting the Med1 locus (guide sequence 5′-TGTCAGGAT-GAAGGCTCAGG-3′) or Med15 (guide sequence 5′ GGACTCGGGCTA-CAAGAGTA-3′). For the pX330 plasmid targeting the RPB1 locus were used a previously published vector (gift from Cornelis Murre, Addgene # 165593)[52]. We constructed knock-in donor vectors that inserted a SNAP-tag or Halo-tag sequence to the N-terminus of the Med1, Med15, or RPB1 coding region, respectively. Vectors contained flanking homology arms of ~800–900 bp. Homology arms were amplified from E14 gDNA. pX330 and knock-in donor plasmids were introduced into mESCs by electroporation using the Neon Transfection System (Thermo Fisher Scientific, MPK10025). Cells were transfected with 400 ng pX330 plasmid and 600 ng donor plasmid per 150 000 cells and electroporated with the following settings: 1400 V, 10 ms pulse width, three pulses. Cells were recovered for 2 days in 2i+LIF media prior to clonal isolation. The SNAP-Med1, and SNAP-Med15 cell lines are homozygous knock-ins while the Halo-Pol2 cell line is a heterozygous knock-in. For visualization of the nucleus in SNAP-Med1 cells, we additionally introduced a pCAGGs-tagBFP2-NLS cassette using the ePiggyBac transposase knock-in vector. TagBFP2 expressing cells were selected using FACS.

### Cloning

The YAP sequence used for all expression constructs represents the mouse isoform that lacks exon 6 as previously reported[7]. All YAP constructs (Halo-YAP, mCherry(Y72F)-Halo-YAP, EGFP-Halo-YAP, GFP(0)-Halo-YAP, GFP(−30)-Halo-YAP, FKBP-Halo-YAP, FKBP-Halo-YAP(WT, bulky charge), FKBP-YAP(well-mixed charge)), control constructs (FKBP-SNAP, FKBP-Halo, tagBFP2-NLS), and components of the SPARK-ON system (Cel-Frb-GFP-NLS-HOTag3, iZF-GFP-NLS-HOTag6) were cloned into the ePiggyBac backbone under control of a pCAGGS promoter using Gibson assembly or restriction enzyme cloning. The non-fluorescent mCherry(Y72F), FKBP sequences, and the parts of the SPARK-ON system were a gift from the Xiaokun Shu lab (UCSF), the supercharged GFPs were a gift from Allie Obermeyer (Addgene #199167)[53].

### Transient transfection of mESCs

Except for the stable ePiggyBac tagBFP2-NLS cell line, all constructs were transiently expressed in mESCs in the YAP KO/SNAP-Med1, YAP KO/Halo-RPB1, or YAP KO background. To transfect mESCs, $1 \times 10^6$ mESCs were electroporated with 3 μg YAP or control expression vector using the Neon Transfection System (Thermo Fisher Scientific, MPK10025). For additional expression of the SPARK-ON components, a total amount of 1 μg iZF-GFP-NLS-HOTag6 expression vector and 3 μg Cel-Frb-GFP-NLS-HOTag3 expression vector was added to the transfection mix. Neon settings for the electroporation were as follows: 1400 V, 10 ms pulse width, three pulses.

### Spontaneous differentiation of mESCs

Prior cell seeding, 96-well glass bottom dishes were coated with 10 μg/ml natural mouse laminin (Thermo Fisher, 23017015) for ~6 h at 37 C. The laminin was removed by pipetting prior cell seeding.

For transiently transfected cells, cells were immediately seeded after electroporation in 100 μl spontaneous differentiation media at ~25 000 cells per well in a 96-well glass bottom plate. Due to cells death from electroporation, this yields a cell density comparable to ~10 000 non-transfected cells per well.

For non-transfected cells, cells were seeded in 100 μl spontaneous differentiation media at ~10 000 cells per well in a 96-well glass bottom plate.

### Mapping of endogenous YAP expression range to transiently transfected YAP expression levels

YAP IF staining was used to compare the endogenous YAP level range of WT mESCs to the expression range of our re-expressed Halo-tagged YAP constructs (expressed in the YAP KO background cell line). Since fixation of the Halo-tag(JFX650) stained cell affects its intensity, we first live imaged the cells to detect their Halo-tag(JFX650) signal to derive a correction factor for the fixation. Then, the cells and WT mESCs (grown under the same conditions) were PFA fixed and YAP levels were compared by YAP immunofluorescence (see section IF staining of mESCs). Nuclear YAP levels were quantified on the WT cells and the 1 % and 99 % percentiles of the population were defined as the lower and upper endogenous expression limits. The IF staining of the transiently transfected cells was used to determine the intensities of the endogenous expression range that corresponds to the corresponding Halo-tag signal. The Halo-tag signal was further corrected for signal loss from fixation. To this end, the mean Halo-YAP(JFX650) levels of the live-imaged and fixed cells was used to determine a correction factor ($\text{corr}_{fix} = \text{YAP}_{live}/\text{YAP}_{fixed}$). To obtain the approximate live imaging Halo signal of live stains, the fixed stains were multiplied by the $\text{corr}_{fix}$.

### Staining with SNAP and Halo-tag ligands

For total protein stains, cells were incubated with 10 nM SNAP or Halo-tag ligand (JFX549 or JFX650) for 30 min in their respective culture media, washed once and incubated for 1 h in culture media prior to further processing.

For single-molecule stains, cells were sparsely labeled by incubation with ~0.05 nM Halo-tag ligand (JFX549) for 30 min in their respective culture media. Cells were counterstained with 10 nM Halo-tag ligand (JFX650) for 10 min to visualize the total YAP expression level. Cells were washed once and incubated for 1 h in culture media prior to further processing.

### IF staining of mESCs

Fixed cells were permeabilized with 0.05 % TritonX-100/ 0.075 % Sodium dodecyl sulfate (Fisher Scientific, BP151-100; Sigma Aldrich, 436143) for 20 min and blocked with 10 % normal goat serum (Abcam, ab7481) for 1 h. Cells were incubated with a 1:100 dilution with YAP primary antibody (Cell Signaling Technology, 14074) in blocking buffer over night at 4 C. Cells were washed three times with 0.01 % TritonX-100 (Fisher Scientific, BP151-100) and incubated with Alexa-488, conjugated secondary antibody (1:1000, Thermo Fisher Scientific) and NucBlue (Thermo Fisher Scientific, R37605) in blocking buffer for 1 h at room temperature. Cells were washed 3 × 15 min with 0.01 % TritonX-100 and incubated in PBS for imaging.

### Epi-illumination selective plane illumination microscopy of YAP dynamics

Epi-illumination selective plane illumination microscopy (eSPIM)[54] was performed on a custom-built setup constructed around an inverted microscope stand (Ti-E, Nikon) equipped with an active focus stabilization system (PFS, Nikon) and a motorized piezo stage (MS-2000, ASI). Samples were illuminated and fluorescence was collected through the same primary objective O1 (CFI Plan Apochromat IR 60x WI NA 1.27, Nikon). The microscope was configured as a dual-function widefield and eSPIM microscope. Widefield imaging was achieved using an LED light source (X-Cite XLED1, Excelitas) on the back port and an sCMOS camera (Orca-Flash 4.0, Hamamatsu) on the left side port of the microscope.

The eSPIM optical path was coupled into the microscope through the right side port. Four illumination lasers (Obis 405, 488, 561, and 640, Coherent) were spectrally filtered using bandpass filters (405/10, 488/10, 561/10, 640/10 nm, Chroma) and combined using a series of dichroic mirrors before being collimated and spatially filtered by a 30 μm pinhole placed between two achromatic lenses (50 mm and 45 mm). The circular beam was elongated along one axis using a pair of achromatic cylindrical lenses (50 mm and 200 mm), clipped by an iris diaphragm placed at a conjugate plane of the primary objective focal plane to control the width of the illumination light sheet, and focused by a third achromatic cylindrical lens (100 mm) to a slit placed at the conjugate plane of the primary objective back focal plane. The slit controls the numerical aperture of the illumination light sheet and was adjusted to optimize the extent and uniformity of out-of-focus excitation reduction across the full thickness of the imaged cell. A translation stage adjusted the offset of the illumination beam to control the tilting angle of the illuminating light sheet so that it matches that of the detection focal plane. The illumination beam is then reflected by a quadband dichroic mirror (ZT405/488/561/650rpc, Chroma) into a pair of relay lenses (TTL100-A and CLS-SL, Thorlabs). A galvanometric mirror (GVS011, Thorlabs) conjugated to the pupil plane of the primary objective, in conjugation with a scan lens (CLS-SL, Thorlabs) at the side port of the microscope, scans the light sheet across the sample. The emitted signal was collected through O1, passed through the internal tube lens (Nikon), and de-scanned using the same scan lens and galvanometric mirror. After the relay lens pair, the emission light is separated from the illumination light by the quad-band dichroic mirror and further filtered by channel-specific bandpass filters (525/50, 605/70, 700/75 nm, Semrock) or a quadband bandpass filter (FF01-440/521/607/700, Semrock) mounted on a motorized filter wheel (FW-103, Thorlabs).

A remote volume with 1.33x overall magnification was formed using two tube lenses (TTL200-A, TTL180-A, Thorlabs) and a 100×0.9 NA secondary objective O2 (U Plan Fluor, Nikon). An oblique plane within the remote volume was imaged using a 'snouty'-type tertiary objective O3 (AMS-AGY v1.0, Applied Scientific Instrumentation)[55] placed at a 30° angle relative to the optical axis of O2. Light collected by O3 was imaged by a tube lens (TTL200-A, Thorlabs) to a back-illuminated sCMOS camera (Prime BSI, Photometrics) with a back-projected pixel size of 122 nm in the sample space. O3 and the camera were placed on a piezo-controlled translation stage for focus adjustment.

Samples were maintained at 37 °C and 5% $CO_2$ using a stage-top incubation chamber with environmental control unit (STXG PLAMX, Tokai Hit). Samples were initially focused in widefield mode before being imaged in light-sheet mode. Single-plane time-lapse data was acquired in static light-sheet mode, i.e. without scanning the light-sheet across the sample. The setup was controlled with Micro-manager 2.0 gamma.

Cells were imaged in spontaneous differentiation media (see above) without phenol red and supplemented with 50 μg/mL ascorbic acid (Sigma Aldrich, A4544) and 1:100 Prolong Live Antifade Reagent (Thermo Fisher, P36975).

For single-molecule imaging (Halo-YAP, sparse labeling with Halo-tag ligand JFX549), cells were images with 50 ms exposure time at 150 ms frame intervals for a total of 500 frames. An additional image of the total YAP stain (Halo-YAP JFX650, full labeling with Halo-tag ligand JFX650) and nuclear marker (tagBFP2-NLS) was taken at the beginning and end of the time series.

For simultaneous imaging of total Halo-YAP protein (full labeling with Halo-tag ligand JFX650) and endogenous SNAP-Med1 condensates (full labeling with SNAP-tag ligand TMR), cells were imaged with 50 ms exposure time for each channel at 250 ms frame intervals for a total of 500 frames. An additional image of the and nuclear

marker (tagBFP2-NLS) was taken at the beginning and end of the time series.

For simultaneous imaging of total SNAP-YAP protein (full labeling with SNAP-tag ligand TMR) and endogenous Halo-Pol2 condensates (full labeling with Halo-tag ligand JFX650), cells were imaged with 50 ms exposure time for each channel at 800 ms frame intervals for a total of 200 frames. The Halo-Pol2 staining was used for nuclear segmentations.

Notably, we found that PFA fixation of our cells interferes with YAP's localization within Med1 condensates. Therefore, we recommend using live imaging to study YAP-Med1 co-condensation.

### Live imaging of synthetic YAP condensates (SPARK-ON system)

The assembly of the functionally inert synthetic condensates (HOTag3, HOTag6) was induced by incubation with 5 μM lenalidomide at ~30 min prior imaging start. YAP or control constructs were recruited to the synthetic condensates by addition of rapamycin to a final concentration of 50 nM during imaging (at ~2 min post-acquisition start). Cells were live imaged at 2 min intervals for a total of 2 h on a Nikon Eclipse Ti inverted confocal microscope (Nikon) equipped with a CSU-W1 Yokogawa spinning disk (Andor), an iXon Ultra EMCCD camera (Andor), and 405, 488, 561, and 640 nm laser lines using a 100× 1.49 NA oil objective (Nikon, pixel size = 0.130 μm). The synthetic condensates, YAP, and Med1 or Pol2 were imaged simultaneously. For the Med1 imaging experiments the additional tagBFP2-NLS signal was imaged in the 405 channel for later segmentation of nuclei.

For inhibition of Pol2 elongation, DRB was added to a final concentration of 100 uM at ~12 min post-acquisition start.

### Quantification of YAP, Med1, and Pol2 recruitment to synthetic condensates

To segment and track nuclei and condensates, the signal from the synthetic condensates (HOTag3/HOTag6) and the nuclei (tagBFP2-NLS, or Pol2 signal) was segmented using a custom trained AI segmentation algorithm form the NIS.ai suite of the NIS-Elements software (Nikon). Then, for each movie, the nuclei and synthetic condensates were tracked using the Fiji Trackmate plugin[56]. For synthetic condensates, only tracks starting in the first frame (t = 0) were included in the quantification. Furthermore, synthetic condensate tracks with track length shorter than 18 min ( = 10 frames) were excluded. We also filtered out any condensates <0.45 a.u. or >1.55 a.u. in area, and only considered condensate tracks that localized to nuclei.

For the quantification of YAP recruitment to the condensates, the mean condensate intensity at t = 0 min was subtracted for each time point of each track. For quantification of the Med1 or Pol2 recruitment, the mean condensate intensities were first background subtracted. Then, to correct for non-specific Med1 or Pol2 recruitment and photobleaching, we quantified the Med1 and Pol2 intensity on condensates upon recruitment of control constructs harboring the fluorophore only (FKBP-Halo-YAP or FKBP-SNAP-YAP). Med1/Pol2 intensity time series from the control were fit with a mono-exponential. The mono-exponential control curve was subtracted from the mean Med1 or Pol2 intensity for each time series of the experimental conditions (FKBP-Halo-YAP, FKBP-SNAP-YAP). Correlation of YAP vs. Med1 or Pol2 recruitment at t = 16 min demonstrated that a minimum of 250 a.u. (Halo-YAP) and 750 a.u. (SNAP-YAP) was required for a significant response of Med1 or Pol2, respectively. Were therefore only considered tracks with YAP recruitment above these values. The data from all experiments was pooled and the mean condensate intensity at t = 0 min was subtracted from all timepoints.

### RNAseq sample preparation and analysis

For RNAseq, YAP KO cells were transfected with plasmids for expression of the SPARK-ON system and FKBP-Halo-YAP as described in sections "Transient Transfection of mESCs" (see above). Cells were

differentiated for 1 d as described in section "mESC culture maintenance and spontaneous differentiation" (see above). Then, to assemble the synthetic condensates (HOTag3, HOTag6), cells were first incubated with 5 μM lenalidomide for 30 min and then incubated with 5uM lenalidomide and 50 nM rapamycin to recruit YAP to the condensates. Non-treated cells (no lenalidomide or rapamycin) served as control. 1 h after rapamycin addition, media was removed, and cells were lysed in Trizol. RNA was extracted according to the Trizol manufacturer's instructions. Following RNA extraction, gDNA was removed using on column digest of the Qiagen RNAeasy kit according to manufacturer instructions. The experiment was performed in three independent experimental repeats. Library preparation and sequencing was performed by the Yale Center for Genome Analysis using poly(A) capture and paired-end sequencing (150 bp) on a NovaSeq X Plus instrument.

Sequencing reads were filtered and trimmed using fastp (version 0.21.0 with --length_required 20 --average_qual 20)[57]. Datasets were aligned against the GRCm39 genome (https://www.ncbi.nlm.nih.gov/datasets/genome/GCF_000001635.27/) using STAR (version 2.7.2a) (--outFilterMismatchNmax 2 --alignIntronMax 15000 --alignMatesGap Max 15000 --outFilterMultimapNmax 100 --winAnchorMultimapNmax 100)[58]. FeatureCounts (version 1.6.4, parameters: -M -C -O)[59] was used to count the reads/fragments over the gene annotations. Differential expression analysis was performed using DESeq2 version 1.26[60,61] on raw read counts to obtain normalized fold changes (FCs) and Padj values for each gene using the statistical models included in the DESeq2 package.

The RNA-seq data were deposited in NCBI SRA and are available under the accession PRJNA1276144.

## Quantification of YAP single-molecule dwell times

YAP single-molecules were quantified from sparsely labeled light-sheet time series. Nuclei were manually segmented using the nuclear tagBFP2-NLS signal. Nuclear YAP single-molecules were segmented and tracked using the Fiji Trackmate plugin[56] with the following settings:

Spot detection: LOG detector, DO_SUBPIXEL_LOCALIZATION: true, RADIUS: 3 pixel, TRESHOLD: 1.2, DO_MEDIAN_FILTERING: false.

Spot tracking: LAP tracker, LINKING_MAX_DISTANCE: 3 pixel, GAP_CLOSING_MAX_DISTANCE: 3 pixel, MAX_FRAME_GAP: 2, ALLOW_TRACK_SPLITTING: false, ALLOW_TRACK_MERGING: false.

The mean nuclear YAP level was quantified from the total YAP stain. We only included nuclei with YAP levels within the endogenous YAP expression range. For each independent experimental repeat, tracks from all nuclei were pooled and the inverse cumulative distribution function was determined.

## Quantification of YAP cluster intensities

YAP cluster intensities were quantified from light sheet movies of fully labeled Halo-YAP cells. Nuclei were manually segmented using the nuclear tagBFP2-NLS signal. Nuclear YAP hubs were segmented and tracked using the Fiji Trackmate plugin with the following settings:

Spot detection: LOG detector, DO_SUBPIXEL_LOCALIZATION: true, RADIUS: 3 pixel, TRESHOLD: 2.5, DO_MEDIAN_FILTERING: false.

Spot tracking: LAP tracker, LINKING_MAX_DISTANCE: 3 pixel, GAP_CLOSING_MAX_DISTANCE: 3 pixel, MAX_FRAME_GAP: 2, ALLOW_TRACK_SPLITTING: false, ALLOW_TRACK_MERGING: false.

To quantify integral YAP cluster intensities, spots coordinates were imported into python (version 3.8.5) and a 2D gaussian (with offset) was fit to the clusters using a custom script (available at https://doi.org/10.5281/zenodo.15683295) that uses code elements from Alexander et al. (2019, Elife)[62]. For each spot, the offset was subtracted and the sum pixel intensity within the area of the gaussian fit was quantified. To estimate the number of molecules per cluster,

we made use of the heterogeneous YAP expression levels and quantified the integral intensity of single YAP molecules in very low YAP expressing cells with ~single-molecule labeling density. To quantify the dense phase YAP intensities per nucleus, we only considered hubs with >=10 YAP molecules and quantified the sum intensities of all clusters per nucleus. To quantify the dilute phase YAP intensity, we excluded the pixel of all YAP clusters and determined the average YAP intensity of the remaining nuclear YAP pixel. The average nuclear YAP levels were quantified as mean nuclear intensity including all pixel.

## IDR sequence analyses

YAP IDR sequences were extracted from the Mus musculus isoform 2 sequence (Uniprot accession P46938-2) using the boundaries of the two WW domains. For Med1, the IDR was extracted from the Mus musculus sequence (Uniprot accession Q925J9) by aligning this sequence with the Homo sapiens sequence (Q15648) and using the Homo sapiens IDR definition from Table 1 from Richter et al.[31]. For Pol2, the IDR was extracted from the Mus musculus sequence (Uniprot accession P08775) using the MobiDB-lite prediction.

The Mus musculus proteome was downloaded from UniProt (UP000000589) and all IDRs greater than 30 amino acids in length were extracted using MobiDB[63,64]. To extract grammar features of YAP, MED1, and Pol2 IDRs, NARDINI+ was performed as described in King et al.[39] except for one change. Here, the mean and standard deviations of each compositional grammar feature were extracted for the Mus musculus IDRome and used to calculate z-scores for the YAP, MED1, and Pol2 IDRs. Although all 90 grammar features were analyzed only those with $|z\text{-score}| >= 1$ for at least one of the IDRs are shown.

## YAP(well-mixed charge) sequence design

The YAP well-mixed sequence was designed by extracting the negative (E, D) and positive (K, R, H) residues within the two IDRs. Negative residues at positions 1, 77, 84, 95, 248, 308, 341, 351, 422, and 454 were maintained at their WT position. We held certain negative residues fixed and added histidine to the list of positive residues to make the number of positive and negative residues approximately equal. Then, charge residues were replaced by swapping the WT charge residue with the next extracted negative residue, followed by the next extracted positive residue, and so on in order to make charged positions mostly every other charge.

## Calculation of YAP charge profiles

To calculate the charge profiles (Fig. S5C, S5D, 6C), we consider residue types pos = {K,R} and neg = {D,E}. For a given sequence, we calculate the fraction of pos residues minus the fraction of neg residues for each sliding window of length 5 (Fig. 6C) or length 31 (Fig. S5C, S5D). Then, the values from all sliding windows that contain a given residue are averaged to yield a residue specific mean net residue type value.

## YAP/Med IDR intermaps

Mpipi intermaps (Figs. S5E, 6D) were calculated using FINCHES[42] using a window size of 31. For Fig. 6D, net charge per window is calculated as the fraction of positive residues minus the fraction of negative residues in each sliding window of size 31. For Fig. S5E, the enrichment of glutamine residues versus all other polar residues is determined by calculating the fraction of {S,T,N,C,H} residues minus the fraction of Q residues in each sliding window of size 31. In Fig. S5D, the mean Mpipi per residue attractive vector is plotted for a window size of 31 for the YAP IDRs.

## YAP surface charge prediction

For the YAP surface charge prediction in Fig. 2A, the YAP sequence was submitted for structure prediction using the AlphaFold3 web

server[65]. We protonated the protein structure and generated electrostatic potential maps using the PDB2PQR pipeline with the PARSE forcefield at a neutral pH in the Adaptive Poisson-Boltzmann Solver online server[66]. We visualized the potential map by projecting it onto a solvent accessible surface representation of the protein in VMD[67].

## Statistics

Details can be found in the legend of each figure. N represents the number of independent experiments. $P$-values $< = 0.05$ were considered statistically significant.

## Reporting summary

Further information on research design is available in the Nature Portfolio Reporting Summary linked to this article.

## Data availability

All data is available upon reasonable request. The RNA-seq data were deposited in NCBI SRA and are available under the accession PRJNA1276144. Source data are provided with this paper.

## Code availability

Custom code for image analysis was deposited in the Zenodo repository and is available at https://doi.org/10.5281/zenodo.15683295.

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

## Acknowledgments

We thank Rohit Pappu for feedback on the manuscript, the entire Weiner lab for helpful discussions, and Axel Poulet for help with RNAseq analysis. This work was supported by the National Institutes of Health R35GM118167 (ODW), U01DK127421 and R01GM131641 (BH), and R01CA258237 (XS). KMR's contributions were supported by a grant from the Air Force Office of Scientific Research (grant FA9550-20-1-0241 to Rohit Pappu). BH is a Chan Zuckerberg Biohub San Francisco Investigator.

## Author contributions

K.M. designed and performed experiments, analyzed and interpreted data, drafted and edited the manuscript. K.Y. designed and performed experiments, analyzed and interpreted data, and edited the manuscript. R.C.-K. designed and performed experiments, analyzed and interpreted data. K.M.R. designed and performed computational theoretical analysis, analyzed and interpreted data, and edited the manuscript. C.-I.C. provided critical resources. X.S. provided critical resources. B.H. supervised the study, interpreted data, designed experiments, drafted and edited the manuscript, acquired funding. O.D.W. supervised the study, interpreted data, designed experiments, drafted and edited the manuscript, acquired funding.

## Competing interests

The authors declare no competing interests.
