## [Transparent Peer Review file · Nature Communications]

YAP charge patterning mediates signal integration through transcriptional co-condensates

Corresponding Author: Professor Orion Weiner

Version 0:

Reviewer comments:

Reviewer #1

(Remarks to the Author)

In this manuscript, Meyer et al. integrate light-sheet single-molecule imaging, synthetic condensates, and IDR sequence analysis to uncover how charge-patterning within YAP's IDRs drives its selective co-condensation with the transcriptional scaffold Med1. Their findings provide key mechanistic insights into the role of YAP condensates in transcriptional feedback regulation, highlighting their potential function as signal integration modules. The study presents a comprehensive and rigorous experimental analysis, but several concerns need to be addressed:

Major concerns:

1. YAP's IDRs are inherently negatively charged, yet the authors introduce a negative charge-tag (Halo-YAP with engineered GFP, net charge -30) to disrupt condensate formation and examine its effect on YAP-chromatin interactions. However, given that DNA is also negatively charged, how do the authors confirm that the observed effects are due to alterations in condensate dynamics rather than a direct impact on YAP-DNA interactions?
2. The light-sheet single-molecule imaging data show weak and transient accumulation of YAP at Med1 and Pol II condensates. However, have the authors validated these findings using co-immunofluorescence (co-IF) experiments in mESCs? Co-IF could provide complementary evidence by assessing whether YAP exhibits colocalization with Med1 and Pol II condensates at steady state, independent of the transient behaviors captured in live imaging.
3. There appear to be inconsistencies in figure citations within the manuscript. Specifically, I could not locate the reference for Fig. 5C and 5D, and there seems to be an incorrect figure reference in the section describing the use of the Pol II inhibitor DRB.
4. The time-course behavior of SNAP-Med1 appears inconsistent between Fig. 4C (wild-type YAP) and Fig. 6E (wild-type/blocky charged YAP). In Fig. 4C, Med1 condensates remain stable or slightly decreased over time upon YAP recruitment, whereas in Fig. 6E, Med1 intensity significantly decreases over time, even in the wild-type YAP condition. What accounts for this discrepancy?

Minor concerns:

1. The font size in some figures is not visually clear, particularly in Fig. 1 and Fig. 6. Improving readability would enhance data interpretation.
2. There appears to be a missing bracket in the following sentence: "These clusters barely exceeded ~30 molecules, and larger clusters were very rare (0.01% of clusters have >30 molecules (Fig. 3C))." The correct formatting should be reviewed to ensure clarity.

Reviewer #2

(Remarks to the Author)

The study presents a cutting-edge single-molecule imaging approach and proposes a compelling model for YAP-mediated transcriptional co-condensation. The findings contribute to our understanding of transcription factor dynamics and their role in signal integration. However, I have the following concerns that should be addressed to strengthen the conclusions:

1. Mediator Complex Components: The Mediator complex is a multi-component assembly, and it remains unclear whether other subunits beyond MED1 tested contribute to co-condensation with YAP. For example, does MED15, which has been implicated in transcriptional co-regulation (PMID: 34789250), play a role in YAP/Mediator interactions? Additional

experiments on the specificity of these interactions would be beneficial.

2. Relevance to Stem Cell Biology: While the study highlights a mechanistic framework for YAP-mediated co-condensates, its functional relevance in stem cell biology needs further clarification. How does this condensate formation impact key stem cell transcriptional programs? Providing direct evidence of its biological significance in stem cell fate decisions would enhance the impact of the study.

3. Coiled-Coil Domain Interactions vs. Electrostatic Interactions: Several studies (PMID: 31792379, PMID: 33606996, PMID: 32203417) have demonstrated that hydrophobic interactions via coiled-coil domains are essential for YAP and TAZ condensation. How do these previously described interactions relate to the charge-mediated mechanism proposed here? A more explicit discussion reconciling these two models would help clarify the broader principles governing YAP condensation.

4. Structural Evidence of Charged Blocks Formation: The study proposes that negatively charged regions in YAP facilitate its interaction with positively charged blocks in MED1. However, YAP is known to be highly dynamic, and its full-length structure has not been resolved. Is there structural evidence—either from experimental techniques such as NMR or from AI-based modeling (e.g., AlphaFold)—to support the formation and dynamics of these charged blocks within IDR-2? Addressing this question would provide stronger mechanistic insights into the proposed model.

Overall, this study presents exciting findings with significant implications for transcriptional regulation through biomolecular condensates. Addressing these concerns would further validate the proposed model and enhance the manuscript's impact.

Reviewer #3

(Remarks to the Author)

Version 1:

Reviewer comments:

Reviewer #1

(Remarks to the Author)

My comments have been addressed.

Reviewer #2

(Remarks to the Author)

The revised version addressed my Concerns. I support to publish in Nat Communications.

Reviewer #3

(Remarks to the Author)

We thank the reviewers for their constructive feedback, which has substantially improved the manuscript. Please find below our point-by-point response.

REVIEWER COMMENTS

Reviewer #1 (Remarks to the Author):

In this manuscript, Meyer et al. integrate light-sheet single-molecule imaging, synthetic condensates, and IDR sequence analysis to uncover how charge-patterning within YAP's IDRs drives its selective co-condensation with the transcriptional scaffold Med1. Their findings provide key mechanistic insights into the role of YAP condensates in transcriptional feedback regulation, highlighting their potential function as signal integration modules. The study presents a comprehensive and rigorous experimental analysis, but several concerns need to be addressed:

We are pleased that the reviewer finds our study comprehensive and rigorous.

Major concerns:

1. YAP's IDRs are inherently negatively charged, yet the authors introduce a negative charge-tag (Halo-YAP with engineered GFP, net charge -30) to disrupt condensate formation and examine its effect on YAP-chromatin interactions. However, given that DNA is also negatively charged, how do the authors confirm that the observed effects are due to alterations in condensate dynamics rather than a direct impact on YAP-DNA interactions?

This is a good point. Our initial perturbations (single-molecule imaging of Halo-YAP fused to an engineered GFP with a net charge of -30, Fig. 2C,D) cannot exclude the possibility that the loss of longer YAP dwell times is the result of repelling forces with the negatively charged DNA backbone. We have now added this caveat to the main text in that context (**lines 114-117**). However, our synthetic condensate system provides two lines of evidence that YAP's charge and/or charge pattern directly controls Med1 condensate dynamics.

First, we demonstrate that the addition of the negatively charged tag to YAP results in a significant loss of Med1 recruitment to the synthetic condensates. Importantly, the rapamycin-inducible system allows us to force the recruitment of the negatively charged tagged YAP to synthetic condensates at levels comparable to the control (**Fig. 4E**, comparable recruitment of YAP is indicated by dashed line). This allows us to overcome potential effects of charge repulsion from DNA.

Second, we demonstrate a loss of YAP-mediated Med1 recruitment to condensates when we disrupt YAP's charge pattern, without altering the protein's net charge (the net charge of both the blocky charged YAP sequence and the well-mixed charge sequence remains the same, as shown in **Fig. 6E**). This shows that YAP facilitates Med1 condensate formation through electrostatic interactions, independent of the DNA backbone charge.

2. The light-sheet single-molecule imaging data show weak and transient accumulation of YAP at Med1 and Pol II condensates. However, have the authors validated these findings using co-immunofluorescence (co-IF) experiments in mESCs? Co-IF could provide complementary evidence by assessing whether YAP exhibits colocalization with Med1 and Pol II condensates at steady state, independent of the transient behaviors captured in live imaging.

We intentionally used live-imaging approaches throughout our study to avoid the artifacts that are commonly observed in fixed condensates (e.g., see PMID: 36795466), and which is a known complication for studying YAP due to its mechanosensitivity. Therefore, to address the reviewers' concern, we first investigated the effect of PFA fixation on YAP-Med1 co-localization by fixing our live imaging reporters (Halo-YAP, SNAP-Med1). Using HiLo imaging, we assessed the presence of YAP at molecular resolution in Med1 spots. Unfortunately, but not unexpectedly, while Med1 condensates are clearly visible, we could not detect YAP/Med1 co-localization in fixed cells (see representative images below). This indicates a fixation artefact, and we suggest avoiding the study of YAP condensates in fixed cells. We have added a note in the methods section to highlight our choice of non-fixed cells (see **lines 846-848**).

Reviewer Fig. 1. Colocalization of Halo-YAP with SNAP-Med1 upon PFA fixation in five representative cells. Quantification of Halo-YAP (green) and SNAP-Med1 (magenta) signals along the yellow lines indicated in the merged images is shown on the right.

3. There appear to be inconsistencies in figure citations within the manuscript. Specifically, I could not locate the reference for Fig. 5C and 5D, and there seems to be an incorrect figure reference in the section describing the use of the Pol II inhibitor DRB.

Thank you for pointing this out. We added references to Fig. 5C and D and corrected the Figure references in the section describing the use of DRB.

4. The time-course behavior of SNAP-Med1 appears inconsistent between Fig. 4C (wild-type YAP) and Fig. 6E (wild-type/blocky charged YAP). In Fig. 4C, Med1 condensates remain stable or slightly decreased over time upon YAP recruitment, whereas in Fig. 6E, Med1 intensity significantly decreases over time, even in the wild-type YAP condition. What accounts for this discrepancy?

It is likely that the decrease is the result of photobleaching (both Med1 quantifications decrease by ~10 a.u.), but the graph in Fig. 6E is more affected because the initial recruitment levels are lower compared to Fig. 4C (Med1 signal is at 23 a.u. after 30 minutes in Fig. 6E, compared to ~45 a.u. after 30 minutes in Fig. 4C). While we don't know the exact reason for this, we observe that the SNAP-TMR ligand staining can be affected by age or freeze/thaw cycles of the stock solution. It is likely that an older batch was used for the experiments in Fig. 6E, making comparable bleaching effects more apparent. Overall, it is difficult to correct for bleaching of these curves, as the stable association of Med1 in condensates leads to less turnover of bleached fluorophores in condensates than in the control, which lacks specific Med1 recruitment and experiences continuous exchange of bleached fluorophores.

Minor concerns:

1. The font size in some figures is not visually clear, particularly in Fig. 1 and Fig. 6. Improving readability would enhance data interpretation.

Thanks for pointing this out. We increased font sizes for Fig. 1 and Fig. 6 (see new Figures below).

Figure 1

Figure 6

2. There appears to be a missing bracket in the following sentence: “These clusters barely exceeded ~30 molecules, and larger clusters were very rare (0.01% of clusters have >30 molecules (Fig. 3C).” The correct formatting should be reviewed to ensure clarity.

Thanks for noticing this. To avoid double brackets, we corrected it to “(0.01% of clusters have >30 molecules, **Fig. 3C**)”

Reviewer #2 (Remarks to the Author):

The study presents a cutting-edge single-molecule imaging approach and proposes a compelling model for YAP-mediated transcriptional co-condensation. The findings contribute to our understanding of transcription factor dynamics and their role in signal integration. However, I have the following concerns that should be addressed to strengthen the conclusions:

We are pleased the reviewer recognizes our work as compelling and cutting-edge.

1. Mediator Complex Components: The Mediator complex is a multi-component assembly, and it remains unclear whether other subunits beyond MED1 tested contribute to co-condensation with YAP. For example, does MED15, which has been implicated in transcriptional co-regulation (PMID: 34789250), play a role in YAP/Mediator interactions? Additional experiments on the specificity of these interactions would be beneficial.

This is a good question. Our work focused on Med1 because it harbored the longest IDR (~1000 nt), but the Mediator complex contains a number of other IDR-containing Mediator subunits that may rely on a similar co-condensation mechanism (see new **Fig. S5A** showing the Mediator complex with IDR containing subunits in color). To address the reviewer’s question, we first tested YAP’s ability to recruit the Med15 subunit to condensates. We labelled the endogenous Med15 subunit with a SNAP tag using a CRISPR/Cas9 knock-in and imaged the recruitment of Med15 using our synthetic condensate assay in mESCs. Following rapamycin-inducible YAP recruitment to synthetic condensates, we observed Med15 recruitment within minutes (see new **Fig. S5B**). The recruitment followed a similar temporal profile as observed for Med1. In other words, sustained YAP recruitment to condensates results in sustained presence of Med15, suggesting that Med15 is also not affected by the negative feedback observed for the transcription machinery (Pol2, see **Fig. 5A,B**). This demonstrates that YAP is sufficient to recruit at least two different Mediator subunits, making it likely that other subunits are also involved.

Next, we probed the role of YAP co-condensation with other IDR-containing subunits, including Med15. To this end we used our computational analysis of their physico-chemical interaction strength. Given that YAP-Med1 co-condense through their electrostatic interactions, we first focused on possible charge-mediated interactions. The new **Fig.S5C** panel shows the charge profiles of all IDR containing Mediator subunits (i.e. Med1, Med14, Med15, Med25, Med26; see new **Fig. S5C**). Compared to the prominent enrichment of charged residues in Med1, the IDR of Med26 is the only other subunit exhibiting a significant enrichment of charged patches (overall positively charged). These could follow

a similar co-condensation mechanism as Med1. By computing the intermolecular interaction parameter ϵ of the Med26 (purple line) IDR for each position along the YAP IDR sequence, we find that the interaction strength is indeed highest in the negative charge patch of YAP's IDR2, suggesting that YAP and Med26 may co-condense through electrostatic interactions, similar to Med1.

Finally, we followed up on the reviewer's request to probe YAP's interaction with Med15. While Med15 does not harbor significantly charged patches that would indicate an interaction with YAP through electrostatic interactions, our intermolecular interaction analysis demonstrates attractive interactions that align with the Q-rich regions of Med15 IDR and polar-rich regions of YAP IDR2 (see new **Fig.S5E**). This suggests that YAP may interact with Med15 through a different mechanism than Med1.

Together, it is possible that YAP co-condenses with multiple Mediator subunits. Specifically, Med26 could follow a similar charge-mediated co-condensation mechanism as Med1.

We now added the new Figure S5 to the Supplement and discuss these results in the main text, **lines 333-360**.

Figure S5 Analysis of intermolecular interactions between YAP and IDR containing Mediator subunits

A) Overview of IDR containing subunits of the Mediator complex according to Richter et al. with Mediator subunit organization as shown in Soutourina et al. Color-coded subunits have IDRs with length ≥ 100 residues. **B)** Time series images of acute YAP recruitment (middle row) to pre-formed synthetic condensates (top row) and the resulting response of the endogenous Med15 protein. YAP was recruited to synthetic condensates at 2 min following rapamycin addition. Green and black dashed circles indicate Med15/YAP double positive condensates. **C)** Analysis of charged residues in IDR containing Mediator subunits. The graphs show the mean net charge per residue using a window size of 31 along the sequence of the indicated Mediator subunit IDR. **D)** Prediction of intermolecular attractions between the Med1 IDR (blue line) or Med26 IDR (purple line) and the YAP IDR sequences (aligned on top). The top graph shows the mean net charge profile along the YAP protein sequence, IDRs are indicated in grey. Bottom graph shows the FINCHES predicted mean attractive interaction parameter ϵ of the Med1 (blue line) or Med26 (purple line) IDR for each position along the YAP IDR sequence. Note the alignment of the predicted interactions with the IDR charge patterns of YAP IDRs (Med1 interactions align with charge blocks of both YAP IDRs; Med26 interactions align with YAP IDR2's negative charge block). **E)** Predicted intermolecular interaction maps for the Med15 IDR and YAP IDR2. Attractive (magenta) and repulsive (green) interaction strengths are predicted based on the chemical physics of the Mpipi forcefield. Graphs on the left and bottom show the enrichment of glutamine (pink) and other polar residues (green) along the IDRs of Med15 and YAP. Note the alignment of the predicted interactions with the Q-rich regions of Med15 IDR and the glutamine and other polar-rich regions of YAP IDR2 (sum of S,T,N,C,H residues; green).

2. Relevance to Stem Cell Biology: While the study highlights a mechanistic framework for YAP-mediated co-condensates, its functional relevance in stem cell biology needs further clarification. How does this condensate formation impact key stem cell transcriptional programs? Providing direct evidence of its biological significance in stem cell fate decisions would enhance the impact of the study.

To probe the functional relevance of YAP-Med1 co-condensates, we analyzed differentially expressed genes by RNA-seq following synthetic YAP condensate formation. Using our synthetic condensate system (SPARK-ON, see **Fig. 4A**), we compared RNA transcripts in cells with rapamycin-induced YAP recruitment to synthetic condensates versus cells with comparable YAP expression but without condensate recruitment. We chose to analyze gene expression at 1 h post YAP condensate recruitment based on our previous observations that YAP-induced RNA Pol 2 recruitment peaks during this window (see **Fig. 5A,B**). Interestingly, despite the presence of Pol 2 at the condensates, the differential gene expression analysis reveals a repressive effect of YAP condensates on gene expression. Out of ~20,000 detected transcripts, 33 are significantly reduced, while only one shows increased levels (new **Fig. S6A, see below**). GO term analysis reveals an enrichment of differentially regulated genes in developmental pathways such as cAMP, TNF, and MAPK signaling (new **Fig. S6B**), suggesting that they could affect mESC fate. While the results demonstrate a lack of mRNA transcript induction, we cannot exclude that the decreased gene expression is a result of the strong recruitment of Pol2 to the condensates at the expense of transcription of other non-condensate regulated genes.

Generally, the lack of a detectable gene expression increase is at odds with the presence of the transcriptional machinery at YAP condensates. This finding is consistent with a previous study that noted the same phenotype (PMID: 38315854) and may point towards a role of YAP in the transcription of regulatory RNAs (e.g., enhancer RNAs) instead of coding transcripts. The multivalent interaction of non-coding RNAs has been shown to suffice to induce and dissolve Med1 phase separation in vitro and is thought to modulate Mediator dynamics. As such, YAP may serve to control Mediator condensate by recruiting the transcription machinery to transcribe regulatory RNAs. This could establish a transcriptionally permissive environment for the integration of other signals. Unraveling this role would require more specialized deep sequencing methods to capture eRNAs and would go beyond the scope of this study. We have now added the new **Fig. S6** to the manuscript and briefly describe these findings in the main text (see **lines 239-253**).

Figure S6 Analysis of gene regulation by synthetic YAP-Med1 co-condensates

A) Volcano plot showing differentially expressed genes regulated by synthetic YAP-Med1 co-condensates. Fold change represents the ratio of gene expression in cells with synthetic condensates at 1 hour post rapamycin-induced YAP recruitment versus cells with synthetic condensates and YAP expression in the absence of rapamycin addition. Significantly upregulated or downregulated transcripts are shown in red or blue, respectively ($|\text{fold change}| \geq 1.5$, $P\text{-value} < 0.05$). **B)** KEGG Pathway analysis of differentially regulated genes in cells with rapamycin-induced synthetic YAP condensates. Count indicates the number of differentially expressed genes in a pathway.

3. Coiled-Coil Domain Interactions vs. Electrostatic Interactions: Several studies (PMID: 31792379, PMID: 33606996, PMID: 32203417) have demonstrated that hydrophobic interactions via coiled-coil domains are essential for YAP and TAZ condensation. How do these previously described interactions relate to the charge-mediated mechanism proposed here? A more explicit discussion reconciling these two models would help clarify the broader principles governing YAP condensation.

We added a paragraph to the discussion section to address this (see **lines 407-413**). In summary, our model system and results differ from previous reports on YAP condensates in two important details. First, in contrast to many other reports (e.g. PMID: 31792379, PMID: 33606996), we do not observe large macromolecular YAP condensates that are detectable by common imaging modalities (e.g. confocal). Instead, YAP molecules are only enriched at small quantities (5-10m molecules) in Med1

condensates in mESCs, suggesting differences in phase behavior between our and other reported systems. Second, the coiled coil domain reported to harbor hydrophobic residues and contributing to YAP condensation is partially missing in our YAP isoform. In mESCs, among the many predicted isoforms, only two YAP isoforms are highly expressed (PMID: 33938099). These isoforms differ in the presence of exon 6 which harbors part of the coiled-coil domain and that was previously reported to drive YAP condensation through hydrophobic interactions (exon 6 residues span position 314-329 coinciding with the position of the coiled-coil as for example indicated in Fig. 1 in PMID: 32981815). Due to importance of the YAP isoform lacking exon 6 for mESC biology (see PMID: 33938099), we are working with this isoform in our study. This may explain why we observe differences in YAP condensation behavior and why electrostatic interactions are dominating YAP-Med1 co-condensation in our study.

4. Structural Evidence of Charged Blocks Formation: The study proposes that negatively charged regions in YAP facilitate its interaction with positively charged blocks in MED1. However, YAP is known to be highly dynamic, and its full-length structure has not been resolved. Is there structural evidence—either from experimental techniques such as NMR or from AI-based modeling (e.g., AlphaFold)—to support the formation and dynamics of these charged blocks within IDR-2? Addressing this question would provide stronger mechanistic insights into the proposed model.

The charge blocks of YAP are located within its IDRs. As the name suggests, these domains do not form a stable structure and cannot be crystallized or predicted with confidence using AlphaFold. The image below shows the AlphaFold prediction, with the colors indicating very low confidence in the predictions within the IDRs (pTM = 0.18). Consequently, making dynamic predictions about how the charged blocks fold or behave is challenging and would go beyond the scope of this study.

Overall, this study presents exciting findings with significant implications for transcriptional regulation through biomolecular condensates. Addressing these concerns would further validate the proposed model and enhance the manuscript's impact.

Reviewer #3 (Remarks to the Author):
